

# Strategies for Comparison of Modern Probabilistic Seismic Hazard Models and Insights from the Germany and France Border Region

Graeme Weatherill[1], Fabrice Cotton[1,4], Guillaume Daniel[2], Irmela Zentner[3], Pablo Iturrieta[1], Christian Bosse[1]

5   [1]GFZ German Research Centre for Geosciences, Potsdam, 14473, Germany
[2]Électricité de France, Aix-en-Provence, 13290, France
[3]Électricité de France, EDF R & D Lab Paris-Saclay, 91120, France
[4]University of Potsdam, Potsdam-Golm, 14476, Germany

*Correspondence to*: graeme.weatherill@gfz-potsdam.de

10   **Abstract.** The latest generation of national and regional probabilistic seismic hazard assessments (PSHA) in Europe presents stakeholders with multiple representations of the hazard in many regions. This raises the question of why and by how much seismic hazard estimates between two or more models differ, not where models overlap geographically but also where new models update existing ones. As modern PSHA incorporates increasingly complex analysis of epistemic uncertainty, the resulting hazard is represented not as a single value or spectrum but rather as probability distribution. Focusing on recent 15 PSHA models for France and Germany, alongside the 2020 European Seismic Hazard Model, we explore the differences in model components and highlight the challenges and strategy for harmonising the different models into a common PSHA calculation software. We then quantify the differences in the source model and seismic hazard probability distributions using metrics based on information theory, illustrating their application to Upper Rhine Graben region. Our analyses reveal the spatial variation and complexity of model differences when viewed as probability distributions and highlight the need for more 20 detailed transparency and replicability of the models when used as a basis for decision making and engineering design.

**Short Summary.** New generations of seismic hazard models are developed with sophisticated approaches to quantify uncertainties in our knowledge of earthquake process. To understand why and how recent state-of-the-art seismic hazard models for France, Germany and Europe differ despite similar underlying assumptions, we present a systematic approach to 25 investigate model-to-model differences and to quantify and visualise them while accounting for their respective uncertainties.

## 1 Introduction

Effective mitigation of seismic risk, be it at a local, national, or regional scale, requires a quantitative assessment not only of the strength or impacts of the perils to which an area may be subject, but also their probability of occurrence over a given time frame. For earthquakes, probabilistic seismic hazard assessment (PSHA) is now established as the primary means through 30 which our understanding of the physical phenomena is translated into a framework that can yield critical information of


relevance for engineering design, urban planning and development, and financial instruments to mitigate the economic impacts of these events on society. Given the volume of information for risk mitigation that PSHA can produce, national and regional scale PSHA models are now available for every country across the globe (Pagani et al, 2020), with many countries now having developed several successive generations of seismic hazard models and, in some regions, multiple models offering different

perspectives on seismic hazard for the same area of interest.

The issue of multiple perspectives on seismic hazard in a region can be an important one to address from the point of view of model developers, but it also has significant implications for the users of the seismic hazard outputs. In the case that a new seismic hazard model for a region is produced that is intended to update or supersede an existing model, while there may be a

recognition that new data for that region and/or developments in PSHA practice justifies revising or updating a seismic hazard model periodically, this revision will inevitably have implications for stakeholders, particularly when hazard is found to increase or decrease substantially at a location as a result of the new information. In Europe, many different countries are confronted with this situation as new generations of national seismic hazard models emerge. There is, however, also a compounding issue, which is the need for Pan-European assessments of seismic hazard. Two major models within the last

decade have resulted from large-scale multi-institution projects that have put a strong focus on incorporating state-of-the-art developments in PSHA, namely the 2013 European Seismic Hazard Model (ESHM13) (Wössner et al., 2015) and the 2020 European Seismic Hazard model (ESHM20) (Danciu et al., 2021).

Since the completion of the ESHM13 many new seismic hazard models have been developed at national scale, among which

are Switzerland (Wiemer et al. 2016), Italy (Meletti et al., 2021), United Kingdom (Mosca et al., 2022), Turkey (Akkar et al., 2018), Germany (Grünthal et al., 2018) and France (Drouet et al., 2020). Several factors have motivated these national scale developments, but chief among these is the establishment Eurocode 8 (EC8; CEN, 2004) as the predominant standard covering earthquake resistant design. EC8 devolves some specific components seismic design requirements to each of the participating member states via their respective National Annexes. Among these components are the seismic hazard map on which the

design levels of seismic input are based. In many cases, national building design authorities have opted to undertake revisions to their national seismic hazard maps, in part aiming to bring these into line with (or even exceeding of) standards for state-of-practice PSHA modelling in Europe set by the ESHM13, but also because new or more detailed data may be available at local scale to allow a refined estimate of hazard that may not be scalable to larger multi-national regions. These national models should form the authoritative reference seismic hazard model for application to engineering design in their respective countries,

though in some cases these models have integrated components or ideas developed within the ESHM13. We also expect this trend to continue with expected updates to Eurocode 8 and following the publication of the ESHM20.

The dual existence of both a regional scale model (or models) and a national model that cover the same territory naturally raises the question of comparison between models. How and why do models differ and how can we quantify differences? It



has become standard practice for modern seismic hazard assessment to contain detailed assessments of epistemic uncertainty
in both the seismogenic source model (SSM) and ground motion model components (GMMs). These are incorporated into the
analysis in the form of logic trees, which generate many seismic hazard curves by enumerating (or sampling) combinations of
alternative models or model parameterisations and their associated weights. Logic trees have been adopted as the standard tool
for epistemic uncertainty assessment in site specific PSHA for several decades, yet at national and/or regional scale the latest

generation of European seismic hazard models is only the second generation to consider epistemic uncertainties as standard
practice. The increased in sophistication and complexity of the logic trees between the first and second generations is
considerable. With a comprehensive treatment of epistemic uncertainty now standard in models, the breadth and definition of
outputs from PSHA means that we cannot quantify differences purely in terms of an increase or decrease in a map of peak
ground acceleration (PGA) with a 10 % PoE in 50 years, but rather we need to consider the differences in terms of distributions

of hazard from the epistemic uncertainty analysis and do so across the range of outputs.

This paper aims to illustrate the full depth of what we mean by "comparison of PSHA models" by focusing on three recent
models that overlap with one another in terms of territory covered but also adopt complex logic trees to quantify epistemic
uncertainty: 1) the PSHA model for metropolitan France by Drouet et al. (2020), 2) the 2016 national seismic hazard model

for Germany prepared by Grünthal et al. (2018) and 3) the 2020 European Seismic Hazard Model (ESHM20). We begin with
a general overview of the three models in section 2, highlighting both the common elements to the models and the critical
differences. As each model has been undertaken using a different PSHA calculation engine we have endeavoured to translate
both the French and German hazard models from their original proprietary software and into the open-source OpenQuake-
engine, which allows us to explore the models in detail, affording us more control over the calculation and understanding the

detailed modelling differences that the PSHA software can introduce. Section 3 will therefore describe the motivations of
translating the models across to another software and some of the lessons learned from this process. With the models
implemented into a common PSHA software we outline various quantitative techniques to explore the differences between
them firstly in terms of the spatial variation in distribution of activity rates (Section 4), and then by looking at the differences
in the hazard outputs for the three models in the France-Germany border region (Section 5). We will conclude with

recommendations on how to approach model to model comparison based on insights gained from our experience. An additional
set of notes has been compiled that expand upon certain topics mentioned in the current paper, which can be found in the
electronic supplementary material.

We hope these recommendations may form a useful reference point for end users of these models when considering how and

why PSHA models for a given region can differ and how to use this information to form a basis for decision making when it
comes to adopting models or migrating from one to another for use in application.





## 2 Overview of the Recent PSHA Models for Europe, France and Germany

The first seismic hazard model we are considering is that of Drouet et al. (2020), which covers all metropolitan France (FR2020 hereafter) and was developed to capitalise on the outcomes of preceding research into seismic hazard emerging from the
SIGMA project (Pecker et al. 2017). New developments included an updated magnitude homogeneous earthquake catalogue (FCAT-17, Manchuel et al., 2017), recently developed ground motion models (GMM) for France (Ameri, 2014; Ameri et al., 2017; Drouet & Cotton, 2015), and refinements to the characterisation of seismic sources and magnitude frequency relations (MFRs) that built on innovative approaches adopted in the Eastern United States (EPRI, 2012). The hazard model is produced assuming a site condition of $V_{S30}$ 800 m/s (Eurocode 8 Class A), with hazard curves calculated at 6,836 sites for PGA and
spectral acceleration periods in the range 0.01 s to 3 s.

The second seismic hazard model considered here is the 2016 national seismic hazard model of Germany (DE2016 hereafter) was prepared by Grünthal et al. (2018) on behalf of the Deutsches Institut für Bautechnik (DIBt) with the aim of providing an up-to-date seismic zonation for the current design code and national annex to Eurocode 8 (DIN 4149). The model covers the
entire national territory of Germany (plus a small band outside the national borders) with hazard curves calculated every 0.1° longitude and latitude, resulting in seismic hazard curves at 6,226 locations across the country for PGA and spectral accelerations for periods between 0.02 s and 3.0 s. As with FR2020, the curves are calculated on a reference site condition of $V_{S30}$ 800 m/s. Hazard maps are produced by 2D interpolation to a finer grid of 0.01° spacing.

The 2020 European Seismic Hazard Model (ESHM20) is the latest generation seismic hazard model for Europe, covering 36 countries from Iceland in the northwest to Turkey in the southeast. As a comprehensive and state-of-the-art multi-national scale model that builds on new data and scientific developments since ESHM13, ESHM20 provides a comprehensive set of seismic hazard curves, hazard maps and uniform hazard spectra calculated at more than 100,000 locations including all Continental Europe, UK and Ireland, Iceland and various islands in the Mediterranean and Atlantic. ESHM20 is not only the
basis for the seismic input parameter maps of $S_\alpha$ and $S_\beta$ that will form an informative annex to the forthcoming Eurocode 8, it also provides the seismic hazard input into the 2020 European Seismic Risk Model for Europe (Crowley et al. 2021). For Eurocode 8, seismic hazard is calculated with respect to the reference soil condition of $V_{S30}$ 800 m/s (assuming depth to the $V_S$ 800 m/s layer of less than 5 m), which is consistent with both FR2020 and DE2016.

Our comparison of the models begins at the level of the model components. At the first level this comprises the *seismogenic source model(s)* and the *ground motion model(s),* but we will subsequently deconstruct the former into elements relating to the delineation of the sources, the calculation and representation of earthquake recurrence in the logic tree. The respective logic trees of our three hazard models (FR2020, DE2016 and ESHM20) all implement *branch sets* to capture epistemic uncertainty on each of these components. An overview of the components of the three models and how they approach the characterisation



of each aspect, and its epistemic uncertainty can be seen in Table 1. The complete logic trees for each of the three models are

seen for FR2020, DE2016 and ESHM20 in Figures 1,2, and 3 respectively.

**Table 1: Comparison of Seismic Hazard Model Components for each of the three models (FR2020, DE2016, ESHM20)**

| Model Component | FR2020 | DE2016 | ESHM20 |
|---|---|---|---|
| Seismogenic Source Model | • Three small-scale area source zonations (SASZ)<br>• One smoothed seismicity (zoneless) model with an adaptive kernel $M_{MAX}$ and $b$-value based on one large-scale area source zonation (LASZ)<br>• No active faults | • Five area source zonations (two LASZ, three SASZ)<br>• Two smoothed seismicity (zoneless) models based on smoothing using Woo (1996) approach – adaptive kernel and fixed-width kernel<br>• Active faults included for the Lower Rhine Graben in Model C | • One SASZ<br>• One combined active fault and smoothed seismicity model with an adaptive kernel<br>• Smoothed seismicity kernel optimised using log-likelihood scoring |
| Magnitude Frequency Relation Calibration | • $a, b$ and $COV(a, b)$ via penalised Maximum Likelihood Estimation (MLE) (EPRI, 2012)<br>• LASZ values used as prior distributions<br>• $M_{MAX}$ distribution using EPRI (2012) methodology | • $a$ and $b$ fit via MLE – depending on number of events in zone (see explanation in Section 2.2)<br>• $M_{MAX}$ distribution using EPRI (2012) methodology<br>• Two MFRs: one fit to all magnitude data, the other to only larger magnitude data | • $a$ and $b$ fit using penalised MLE with LASZ used for prior distribution<br>• $M_{MAX}$ based on three values (originally shaped on posterior distribution from EPRI methodology): $M_{MAX}^{obs}$, $M_{MAX}^{obs} + 0.3$ and $M_{MAX}^{obs} + 0.6$ |
| Magnitude Frequency Relation Logic Tree | • $a$ and $b$ sampled from multivariate Gaussian – each a separate branch<br>• Stratified sampling (see Appendix #)<br>• $M_{MAX}$ sampled from posterior distribution – with stratified sampling independent of $a$ and $b$<br>• 100 branches (1 per sample) | • Posterior distribution of $M_{MAX}$ discretised into 5 branches (Miller & Rice, 1983)<br>• Activity rates determined from $COV(a, b)$ for each $M_{MAX}$ branch, discretised into 4 branches according to Stromeyer & Grünthal (2015) Appendix B<br>• 40 branches in total | • For each $M_{MAX}$, $COV(a, b)$ is randomly sampled and the 16th, 50th and 84th percentile activity rates used for each magnitude<br>• Two MFRs: 1) truncated Gutenberg-Richter, 2) tapered Pareto<br>• For active fault sources, include uncertainty on $b$-value, slip rate and $M_{MAX}$ |
| Ground Motion Model | • Four GMMs with equal weights: Ameri (2014); Abrahamson et al. (2014); Cauzzi et al. (2015);Drouet & Cotton (2015)<br>• Represents local (France)[Am14, DC15] and "global" [ASK14, C15]<br>• 0.5 weight on "local", 0.5 on "global | • 5 GMMs: Akkar et al. (2014a) [Ak14]; Bindi et al. (2014) [Bi14]; Derras et al. (2014) [De14]; Cauzzi et al. (2014) [C15]; Bindi et al. (2017) [Bi17]<br>• Weights split evenly between European models (Ak14, Bi14, De14), and "global" models (Bi17, C15)<br>• 4 branches with additional stress drop scaling | • Regionalized scaled backbone GMM (Kotha et al., 2020; Weatherill et al. 2020)<br>• 5 branches for stress parameter scaling, and 3 for residual attenuation scaling<br>• Branch weights based on uncertainty distributions (Miller & Rice, 1983)<br>• Calibrated to local data, where available |
| Branches | 1600 | 4040 | 315 (West) / 5985 (East) |




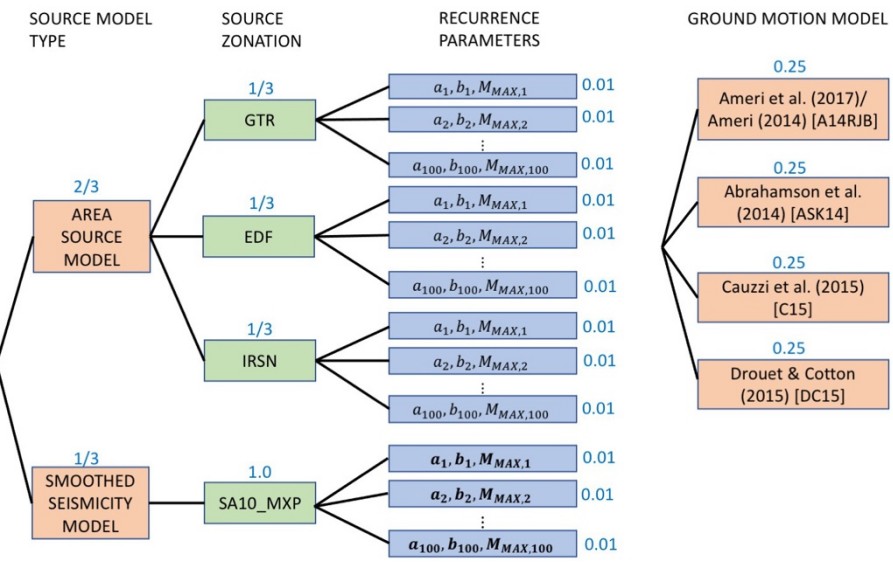


**Figure 1. Complete logic tree for France (Drouet et al., 2020) containing both the seismogenic source model and ground motion model**

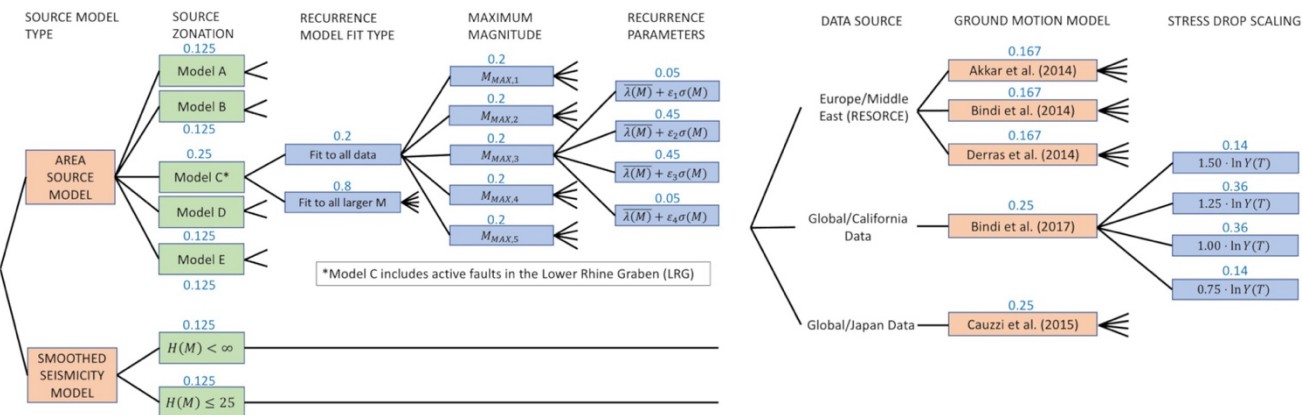

**Figure 2. Complete logic tree of seismogenic source models (left) and ground motion models (right) for DE2016 (Grünthal et al.**
**2018)**

## 2.1 Representation of the Seismic Source

As our focus is on Germany/France, we are working in areas of primarily low-to-moderate seismicity and low tectonic deformation. Although active faults have been mapped in certain areas, most notably the Lower Rhine Graben (Vanneste et al., 2013), not all the assessment have aimed to represent these explicitly in the seismic source models, or they have only
chosen to do so in some branches. As such, each set of seismogenic source models comprise principally *area source zones* and/or gridded seismicity *zoneless* sources. These types of sources are known as *distributed seismicity* sources, and earthquake recurrence is modelled mostly by a double-truncated Gutenberg-Richter model whose parameters $a$, $b$, $COV(a,b)$ and


maximum magnitude ($M_{MAX}$) are constrained by fit to observed seismicity in each zone. The area zonations of the three models can be found in Electronic Supplement A: Note 1.


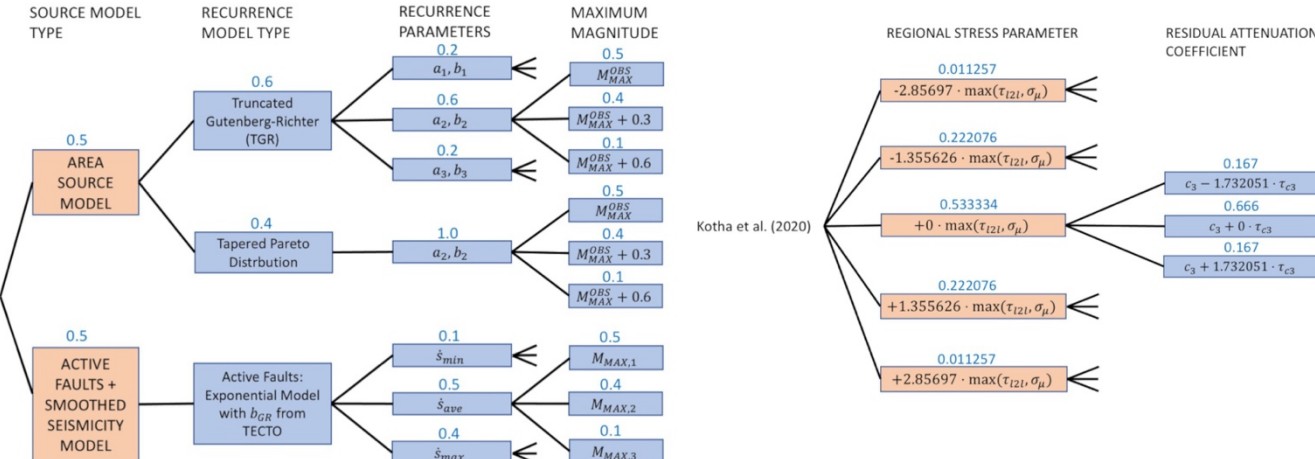

**Figure 3. Complete logic tree of seismogenic source models (left) and ground motion models (right) for ESHM20 (Danciu et al. 2021)**

FR2020 adopts three area source zonations, which assimilate those implemented in previous studies by different orgnisations:

Geoter (now Fugro) (GTR), Électricité de France (EDF) and Institut de Radioprotection et de Sûreté Nucléaire (IRSN). In addition, a single zoneless source model branch is included, which is developed using smoothed seismicity with an adaptive kernel bandwidth applied to the observed seismicity in France from 1960 to 2017. The smoothed seismicity produces seismic sources in the form of 10 km × 10 km cells, with activity rate (a-value) varying cell-by-cell but b-value and $M_{MAX}$ calculated based on the location of the cell with respect to a set of superzones; large scale area zones delineating tectonically based

domains ("Grands Domaines").

DE2016 adopts a similar approach to that of FR2020 by using five alternative area source zonations (Models A, B, C, D and E) alongside two zoneless smoothed seismicity models. For the area sources, Grünthal et al. (2018) explicitly formulate their logic tree as a combination of large-scale area source zones (LASZ) and small-scale area source zones (SASZ). Models A and

B are LASZ and are predicated on the assumption that the regional-scale tectonics are the main factors delineating the seismic sources and that seismicity may be uniform across large areas when viewed at longer timescales than those captured by the observed seismicity. Models C to E are SASZ, which consider local scale seismicity and geological features as the primary guide to the seismogenic sources and therefore delineate smaller scale zones. The smoothed seismicity branches differ in approach from those found in both FR2020 and ESHM20, as DE2016 uses an adaptive kernel with magnitude-dependent

bandwidth based on the method of Woo (1996). The two branches are equally weighted and consider the two cases in which the bandwidth is capped at 25 km ($H(m) \leq 25$ km) and one in which it is unconstrained ($H(m) \leq \infty$). One feature of note in



the area sources are that SASZ Model C adds explicit active fault sources in the Lower Rhine Graben (LRG). These adopt the fault geometry proposed by Vanneste et al. (2013) but used observed seismicity with M ≥ 5.3 across two catchments (area sources) to constrain long-term activity rates for the faults. The activity rates for M ≥ 5.3 within the two catchments are
distributed among the faults within the catchments according to their respective fault length, while for M < 5.3 the catchments are treated as area source. This combined area and fault source model receives the highest weighting of the five source models.

The seismogenic source model of ESHM20 follows a different approach to either that of FR2020 or DE2016. In terms of the number of different source models considered, the source model branch set is simpler. It contains one branch of exclusively
area source zones and another branch for a combined smoothed seismicity and active fault model. As described in Danciu et al. (2021) the area source model aims to unify existing area source zonations from different national PSHA models across Europe, modifying the source geometries at the boundaries of models to ensure a seamless transition from region one to another. In the France/Germany region, the unified area source model adopts DE2016's Model C as its basis in Germany and the IRSN source model branch of FR2020 for France, alongside existing models from Belgium, Switzerland, and the United
Kingdom. The active fault and smoothed seismicity model includes explicit characterisation of faults in both the Upper and Lower Rhine Graben regions, as well as numerous faults in France adapted from the data set of Jomard et al. (2017). Smoothed seismicity is characterised using an isotropic power law kernel with adaptive bandwidth, whose parameters are optimised using log-likelihood scoring (Nandan et al. 2022). To combine the smoothed seismicity with the active faults, a buffer zone is defined for each fault, within which magnitudes lower than a fault-size dependent threshold are kept as smoothed seismicity, while
magnitudes larger than the threshold as associated to the fault surface. For regions away from the fault, b-value and $M_{MAX}$ are based those fit to sources in a large-scale zonation, reflecting regional scale tectonics (named TECTO). More information on the relevance of this will be seen in section 2.2.

In this first component we can see that the three PSHA models display both similarities and differences in their approach to
characterising epistemic uncertainty in the seismogenic source model. FR2020 and DE2016 aim to represent uncertainty in the sources predominantly through multiple uniform area zonations, while ESHM20 divides its weights more evenly between two different source typology definitions. Though only DE2016 adopts explicitly the LASZ/SASZ characterisation, this same philosophy is present in FR2020's "Grands Domaines" model and ESHM20's TECTO model. In the FR2020 model the distinction between large- and small-scale zone models within the three zonations considered (GTR, EDF and IRSN) is not
quite so clear and intentional as it had been for DE2016. Where the contrast exists, it manifests mostly in the difference between the IRSN and EDF models (46 and 49 zones respectively) and the zonation provided by GTR (92 zones). Each of these three models could be described as delineating zones accounting both for geology and seismology, albeit in proportions that are difficult to define. Only DE2016 models the LASZ explicitly in its A and B source zonations; however, all three models will come to adopt similar approaches toward earthquake recurrence by using their LASZ as a basis for fitting their earthquake





recurrence models, which may then inform (either by direct calibration or as a prior distribution) the MFRs for the small-scale area sources with few events. In that sense, the philosophies toward area zonation are similar, but their implementation differs.

Adaptive kernel smoothed seismicity source models are present in all respective logic trees, though each PSHA model has taken a different approach to characterisation and implementation. Both FR2020 and ESHM20 have used approaches similar

to that of, for example, Helmstetter & Werner (2012), optimising the parameters controlling the adaptive kernel's bandwidth using log-likelihood analysis applied to a pseudo-prospective seismicity forecast. The models arrive a significantly different outcomes in terms of the spatial distribution of activity rate, however. DE2016 adopts a different approach by using magnitude-dependent adaptive kernels, which increase the bandwidth for larger magnitudes meaning that the rate in many low seismicity regions is dominated by activity from the most extreme events. This contrasts with the adaptive bandwidth methods used in

FR2020 and ESHM20 for which the bandwidth is based on the density of seismicity. For FR2020 and DE2016 the total weight assigned to the smoothed seismicity branches is the same (0.25), while for ESHM20 the smoothed seismicity/active faults branch receives half the total weight.

**2.2 Magnitude Frequency Relation (MFR)**

For the majority of the seismic sources found within the three source model logic trees (DE2016, FR2020, ESHM20) a

truncated Gutenberg-Richter model is assumed. The only exceptions to this are the DE2016 smoothed seismicity models (which may be considered non-parametric recurrence models) and those branches of the ESHM20 for which a tapered Pareto model is used. In all three regional seismic hazard models, epistemic uncertainty on the recurrence model is included, both in terms of its $a$- and $b$-value as well as $M_{MAX}$.

The general form of the truncated Gutenberg-Richter model to determine the rate $v(M)$ of earthquakes with magnitude greater than or equal to $M$ is:

$$v(M) = v_0 \int_M^{M_{MAX}} \frac{\beta e^{-\beta m}}{e^{-\beta M_{MIN}} - e^{-\beta M_{MAX}}} \, dm \tag{1}$$

where $\beta = b \ln(10)$ and $v_0$ is the rate of earthquakes greater than or equal to minimum magnitude $M_{MIN}$, which can be retrieved from the a-value by $v_0 = \frac{e^\alpha}{\beta} (e^{-\beta M_{MIN}} - e^{-\beta M_{MAX}})$ where $\alpha = a \ln(10)$. As both France and Germany are regions

that would be characterized as low-to-moderate seismicity, the number of events per individual source zone is often too small to determine $a$ and $b$. All three models address this issue in a similar way by invoking the concept of large scale *superzones* that span a sufficiently large region from which to define estimates of the recurrence parameters using a maximum likelihood estimator accounting for the temporal variation in catalogue completeness (Weichert, 1980). The $a$ and $b$ values from these superzones then act as prior distributions for estimates of each source zone in the respective seismogenic source models within

a penalized maximum likelihood estimation (MLE) approach (FR2020), or alternatively maximising a likelihood function assuming a common $b$ value across multiple zones but with seismicity rate varying for each zone (described in Appendix B of





Stromeyer & Grünthal, 2015). For specific details of how the two approaches perform the MLE and how they account for uncertainties in the catalogue and its completeness, the reader is referred to the original publications. The relevant point here is that either approach will define for each source zone an expected $\hat{a}$- and $\hat{b}$-value (or similarly $\hat{\alpha}$ and $\hat{\beta}$) and corresponding

covariance matrix $\boldsymbol{COV}(\alpha, \beta)$ from which we retrieve the uncertainties $\sigma_\alpha$ and $\sigma_\beta$ and their correlation $\rho_{\sigma_\alpha, \sigma_\beta}$. Where individual source zones contain very few events, or span an insufficiently wide magnitude range, the distributions of the recurrence parameters may be informed by, or be fit according to, the superzone to which the source zone is assigned.

The superzone concept is critical for each of the models, not only in defining estimates of $a$ and $b$ value, but also for

characterization of $M_{MAX}$. Here both the FR2020 and DE2016 adopt the EPRI methodology to characterize the distribution $M_{MAX}$ (Johnston et al, 1994; EPRI, 2012); a Bayesian approach in which a global prior Gaussian distribution of $M_{MAX}$ is defined based on the observed maximum magnitudes in analogous tectonically stable regions across the Earth, which is then updated for each superzone such that $f(M_{MAX}) = 0$ for $M_{MAX} < M_{MAX}^{obs}$ in any given region and the posterior distribution combines the shape of the prior and corresponding likelihood function $\mathcal{L}(M|\beta, N_{EQ})$. $\mathcal{L}$ is dependent both on the $b$ value of the

zone as well as the number of earthquakes observed during the corresponding period. The resulting posterior distribution is either sampled (in the case of FR2020) or approximated by a discrete set of weighted values using Miller & Rice (1983) (in the case of DE2016). ESHM20 updates an earlier work of Meletti et al. (2013) to define the $M_{MAX}$ distribution, which yields the three branches $M_{MAX}^{obs}, M_{MAX}^{obs} + 0.3, M_{MAX}^{obs} + 0.6$ assigned weights of 0.5, 0.4 and 0.1 respectively. Though not explicitly applying the EPRI methodology, the weights assigned to each of the three branches reflect an interpretation of a posterior

distribution for $f(M_{MAX})$ that is broadly consistent with those of the EPRI approach.

As the superzones are acting as larger-scale constraints on the parameters of the MFR ($a$, $b$ and $M_{MAX}$) for regions of tectonic similarity, it is inevitable that their definition is based almost exclusively on tectonic and geological criteria rather than local scale seismicity. This is applied consistently across all three models: the "Grands Domaines" for FR2020, LASZ Model A for

DE2016, and the TECTO model for ESHM20. The three superzonations are compared in Figure 4. Interesting to note here, however, is that in the regions where these models overlap there is a considerable degree of divergence in the tectonic zonations. ESHM20 and DE2016 are perhaps more consistent with one another in defining three zones of similar extent that define the Paris Basin, the Upper Rhine Graben and the South German Block. In the lower Rhine Graben and continuing through the Low Countries and into the North Sea, however, all three models diverge. Though far from the only factor that

will eventually contribute toward the differences between the three models in terms of seismic hazard, this divergence in the tectonic interpretations in the superzone models will inevitably propagate into the recurrence models, particularly in regions of low seismicity where the superzones act to fix parameters of, or provide strong priors for, the resulting MFRs.


Though we have so far focused the attention on the definition of the superzones and their influence in constraining the MFRs

themselves, equally important in terms of the impact on PSHA is how the resulting distribution of $\hat{a}$, $\hat{b}$ and $\boldsymbol{COV}(a, b)$ (or

$\boldsymbol{COV}(\alpha, \beta)$) are evaluated within the logic tree. Here, there is yet again significant divergence between the models, with each

model constructing the logic tree for MFR epistemic uncertainty using an entirely different approach.

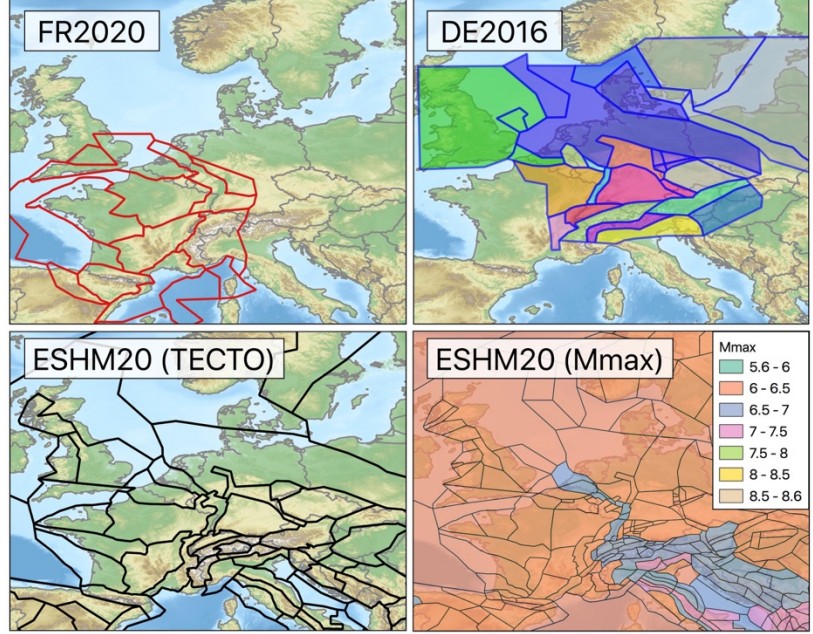

**Figure 4: Large-scale area source zonations (LASZ) assumed for FR2020 (top left), DE2016 (top right) and the two LASZ**
**zonations for ESHM20 based on regional tectonics (bottom left) and maximum magnitude (bottom right). Colours for DE2016**
        **indicate the groupings of LASZ (from Model A) sharing a common b-value.**

DE2016 follows the methodology set out by Stromeyer & Grünthal (2015), who describe the uncertainty in cumulative activity

rate $\nu$ at each magnitude $m$, from the covariance matrix such that:

$\boldsymbol{\sigma}^2(m) = \begin{pmatrix} 1 \\ -m \end{pmatrix}^T \boldsymbol{COV}(a, b) \begin{pmatrix} 1 \\ -m \end{pmatrix} = \sigma_a^2 - 2m\sigma_a\sigma_b + m^2\sigma_b^2$        (2)

The cumulative rate of events greater than or equal to magnitude $\nu_c(m)$ then becomes:

$\nu_{c,i}(m) = \int_M^{M_{MAX}} 10^{\overline{a} - \overline{b}m + \sigma(m)z_i} \, dm$        (3)

where $z_i$ is the number of standard deviations of a standard normal distribution. The incremental activity rate in any given bin

of with $dm$ then simply becomes $10^{\overline{a} - \overline{b}m + \sigma(m)z_i}$. The uncertainty on each magnitude is now represented by a marginal

distribution of $\mathcal{N}(0, \sigma(m))$ the epistemic uncertainty can be mapped into a discrete set of $i = 1, 2, \ldots k$ branches such that $z_i$

and its corresponding weight, $w_i$, are discrete approximations to the standard normal distribution via the Gaussian Quadrature

approach of Miller and Rice (1983). As equation 1 is dependent on $M_{MAX}$, the posterior distribution $f(M_{MAX})$ returned by the

EPRI approach for each zone is first approximated into five discrete branches using the same Gaussian Quadrature method.


Each of the five $M_{MAX}$ values are then input into (3), which is then discretised into four branches to approximate $\mathcal{N}(0, \sigma(m))$.
The epistemic uncertainty in MFR for each area source is therefore represented by 20 logic tree branches (shown in Figure 2).

ESHM20 starts from a similar point as DE2016, as it defines $\hat{a}$, $\hat{b}$ and $\mathbf{COV}(a, b)$ according to Stromeyer & Grünthal (2015) but then approximates the distribution differently. Monte Carlo sampling is used to generate 1 million realisations of $a$ and $b$ from the multivariate normal distribution, and from these samples the pairs corresponding to the 16th, 50th and 84th percentile
values are taken to define the lower, middle and upper branches respectively, with weights of 0.2, 0.6 and 0.2 respectively. $M_{MAX}$ is defined independently of $a$ and $b$ using the three branches described previously. Though the ESHM20 evaluates the multivariate distribution of $a$, b and $\mathbf{C}(a, b)$ in a slightly less formally correct manner compared to that of DE2016, one would still expect the distribution of resulting hazard curves to be similar. ESHM20 diverges further from both the DE2016 and FR2020 approaches, however, by introducing as an alternative set of MFR branches a tapered Gutenberg-Richter recurrence
model (Kagen, 2002):

$$v(M_0) = \left(\frac{M_t}{M_0}\right)^\beta \exp\left(\frac{M_t - M_0}{M_{cm}}\right) \qquad \text{for} \quad M_t \leq M_0 < \infty \qquad (4)$$

where $M_0$ is the seismic moment of an event with magnitude $m$, $M_t$ the threshold moment, $\beta = b\ln(10)$ and $M_{cm}$ the corner moment. Unlike the truncated Gutenberg-Richter model, the tapered Gutenberg-Richter distribution is theoretically unbounded at large moments; however, the exponential decay in the functional form above $M_{cm}$ effectively tapers the rate of events to
triviality for magnitudes larger than the corresponding $M_{cm}$, so truncation can be safely applied within $0.2 - 0.3$ magnitude units above $M_{cm}$ with only minimal impact on the hazard calculation. For the set of branches corresponding to this distribution the rate and b-value are fixed according to the $\hat{a}$ and $\hat{b}$ values defined previously, while the three $M_{MAX}$ branches are applied as epistemic uncertainty on $M_{cm}$. In total, for area sources the source model logic tree contains 12 branches to represent the uncertainty in the MFR: for the truncated Gutenberg-Richter model three branches of $a$ and $b$ and another three of $M_{MAX}$, and
for the tapered Gutenberg-Richter model only three branches for $M_{cm}$.

For both DE2016 and ESHM20 it is also necessary to define activity rates for both the smoothed seismicity sources and the active fault sources. Because of its implicitly non-parametric approach to defining activity rates, no MFR uncertainty is considered for the zoneless smoothed seismicity model of DE2016. Similarly, for ESHM20 the smoothed seismicity model is
optimized through an iterative forecast testing approach, which yields a single preferred smoothed seismicity model without epistemic uncertainty on the MFR. Both models do define epistemic uncertainty on the activity rates for the fault-based models. In the case of DE2016 the maximum magnitudes on the composite fault sources are characterized according to their fault dimension using a normal distribution of $\mathcal{N}(M_{MAX}, 0.3)$ (Vanneste et al. 2013). These distributions are mapped into 5 branches using the Miller and Rice (1983) methodology. On-fault recurrence is modelled using a truncated Gutenberg-Richter relation,
but as the authors could not constrain the proportion of aseismic slip they opted to assign the seismicity for $M_W \geq 5.3$ to the fault sources and the rest to their respective catchment zone (Model C, zones C15 and C22), with the proportion of seismicity





rate for each fault assigned according to the relative length of the fault. This results in a total of 20 MFR branches on the fault sources, comprising five $M_{MAX}$ branches and the four branches of recurrence uncertainty from the catchment zones. In ESHM20 the recurrence models for the active fault sources also use a truncated Gutenberg-Richter model, albeit moment
balanced from the geological coseismic slip rate. As the slip rates are themselves uncertain, three branches for alternative co-seismic slip rates are considered along with three branches for $M_{MAX}$.

FR2020 takes a different approach to characterizing epistemic uncertainty than either ESHM20 or DE2016. For each area source and for each larger-scale superzone the seismicity is represented by a truncated Gutenberg-Richter model represented
by $\hat{a}$, $\hat{b}$ and $\boldsymbol{C}(a,b)$, in addition to the posterior density function $f(M_{MAX})$ that is defined for each superzone. Rather than discretise the distributions of $v(m)$ (as DE2016) or of $a$, $b$ and $M_{MAX}$ into a small set of branches according to Miller and Rice, (1983), Drouet et al. (2020) instead use Monte Carlo sampling, drawing 100 samples from each distribution with each sample then represented as an equally weighted MFR branch (weight = 1 / 100) in the logic tree. Samples are drawn independently from $f(M_{MAX})$ and from the multivariate normal distribution representing the $a$ and $b$ values $MVN\left(\begin{smallmatrix} \hat{a} \\ \hat{b} \end{smallmatrix}, \boldsymbol{C}(a,b)\right)$. This results
in a total of 400 source model branches from four source models (GTR, EDF, IRSN and Zoneless), each with 100 MFR samples. Implementation of the model revealed that the original authors had adopted a stratified sampling strategy for $a$ and $b$, which is illustrated in more detail in the Electronic Supplement Appendix A: Note 2.

**2.3 Upper Rhine Graben Source Example: Similar Approaches, Different Outcomes**

To illustrate how the different approaches to characterization and implementation of the MFRs in a logic tree can yield quite different distributions of activity rate for a given source, even where many inputs to the source model are similar, we consider the case of the Upper Rhine Graben (URG). Among the different source zonations within the different logic trees there are some differences to the exact shape of the source(s) in the Upper Rhine, though most models describe a source that follows
the main outline of the graben starting just north of the Basel earthquake sequence in the south and terminating close to Karlsruhe in the northwest. We select the zone DEAS107 from the ESHM20 unified area source model branch, the FRS zone from the FR2020 GTR source zonation and the D051 zone from the DE2016 model to look at in detail as they depict similar geometries with respect to the spatial seismicity distribution. These sources are shown with seismicity from their respective earthquake catalogues in the top row of Figure 5. Here we observe a first point of divergence, as the catalogues show
remarkably different patterns of seismicity for the same zone. This is somewhat surprising as the ESHM20 adopts the same F-CAT earthquake catalogue as FR2020 within the French territory and the same DE2016 catalogue within the German territory for the post-1900 seismicity. Differences emerge in the pre-1900 earthquake catalogues as ESHM20 adopts the European Pre-Instrumental Earthquake Catalogue (EPICA) (Rovida et al., 2022), which is compiled independently to the other catalogues.



**Figure 5: Example comparison of fit and representation of earthquake recurrence for the Upper Rhine Graben (URG) for the ESHM20 (left column), FR2020 (middle column) and DE2016 (right column). Example geometry of the selected URG seismic source in different models (top row), distribution magnitude with time for the respective zones and the corresponding temporal completeness magnitude assumed by the model (middle row), and distribution of magnitude frequency relations for the zone colour scaled according to weight.**

The next point of divergence can be seen in the estimate of completeness magnitude and its variation in time, which can be seen in the middle row of Figure 5 and given in Table 2. FR2020 and DE2016 estimate completeness using the method of Hakimhashemi & Grünthal (2012), albeit adopting different spatial zones to apply the method, while ESHM20 estimates completeness using an inversion method based on forecast testing (Nadam et al. 2022). Drouet et al. (2020) provide the uncertainty range for the completeness estimates, and although the preferred values are different for many magnitude bins, the





earliest years of completeness for magnitudes in the range $4.0 \leq M_W \leq 6.5$ for the DE2016 and ESHM20 models are consistent with the uncertainty range shown in Table 2 for FR2020. Taking the best estimates, however, and contrasting these against the catalogues (shown in the middle row of Figure 5), it is obvious that both the catalogues and completeness estimates are

dissimilar.

The bottom row of Figure 5 shows the distributions of activity rate with magnitude for all the MFR branches assumed by the respective logic trees. Although each model is using some form of maximum likelihood estimate (Johnson et al., 1994; Stromeyer & Grünthal, 2015) to determine the Gutenberg-Richter parameters for the zone, the results are significantly

different. ESHM20 finds an expected $a$ and $b$ value of 1.9565 and 0.7334 respectively, which are mapped into three branches of $a, b$ pairs: (1.886, 0.685), (1.9565, 0.7443), (2.0278, 0.803). By contrast FR2020 returns $a$ and $b$ values of $2.3711 \pm 0.182$ and $0.8696 \pm 0.0918$ respectively, with $\rho_{ab} = 0.8991$, and while DE2016 is dependent on $M_{MAX}$ the $a$ and $b$ values range from 3.89 to 2.86 and from 1.08 to 0.95 respectively. Not only do the MFR parameters themselves vary then significantly, but Figure 5 illustrates how the different mappings into logic tree branches yield significantly different activity rate distributions.

ESHM20 places more weight on the middle branches, and in this case the MFR logic tree mixes both the truncated Gutenberg-Richter and the tapered Pareto distributions. FR2020 clearly shows the largest spread of MFRs, which arises in part from the independence of $a$ and $b$ from $M_{MAX}$ and in part because of the large number of evenly weighted sample values. DE2016 is something of a middle point, with a narrower range of values and notably higher weights on a specific sub-set of branches.

**Table 2: Variation in completeness window for each magnitude bin assumed for the selected URG source zone**

| Magnitude Bin | FR2020 | DE2016 | ESHM20 |
|---|---|---|---|
| 2.5 – 3.0 | 1970 [1965 – 1975] | 1973/74 | - |
| 3.0 – 3.5 | 1950 [1940 – 1960] | 1870 | - |
| 3.5 – 4.0 | 1850 [1800 – 1875] | 1870 | 1857 |
| 4.0 –4.5 | 1850 [1800 – 1875] | 1870 | 1822 |
| 4.5 – 5.0 | 1700 [1650 – 1800] | 1800 | 1822 |
| 5.0 – 5.5 | 1600 [1500 – 1700] | 1650 | 1479 |
| 5.5 – 6.0 | 1500 [1400 – 1600] | 1450 | 1479 |
| 6.0 – 6.5 | 1500 [1400 – 1600] | 1250 | 1479 |
| $\geq 6.5$ | 1500 [1400 – 1600] | 1250 | 1479 |

The comparison here is not an exhaustive description of all the reasons for what we will eventually see as the differences in seismic hazard between the three models, but it is illustrative of how they can diverge significantly in the critical information for PSHA (namely activity rate per magnitude bin) despite adopting theoretically similar approaches. Particularly insightful is

the contrast in the way in which the distribution of $a$ and $b$ is mapped into the epistemic uncertainty, which would potentially suggest that even if the three models produced a similar fit in their recurrence models, they could still diverge significantly in the resulting activity rate distributions inside the PSHA calculation. We will discuss in the conclusions chapter the implications here for future harmonization of the seismic hazard, but a key point to take from this brief analysis is that each step of the





process from the basic earthquake data through to the distribution of activities rates requires both transparency and scrutiny.
Though the models considered here are arguably better documented than many, there are still many steps in the processes that are not completely described, or if they are described it may be difficult to perceive how this can influence the hazard. These factors will contribute to the differences in hazard model components and hazard model outputs shown in sections 4 and 5.

**2.4 Ground Motion Models**

For the ground motion model (GMM) logic tree it is not necessarily the technical process itself, and the decisions made therein,
that differs significantly between the three PSHA models, but rather the general philosophy of how to characterize epistemic uncertainty. Specifically, between the three models we see an example of a multi-model (or "weights-on-models") GMM logic tree (FR2020) a hybrid multi-model logic tree with backbone scaling factors (DE2016) and a fully regionalized scaled backbone logic tree (ESHM20). All three models explicitly invoke the same objective of "capturing epistemic uncertainty in terms of the centre, body and range of the technically-defensible interpretations of available data" (USNRC, 2012). To contrast
distributions of GMMs from different PSHA models we have created a set of trellis plots, in which the GMM selections from two different models are plotted side-by-side for the same set of predictor variables. The range of GMM median or standard deviation values for the contrasting model is described by a shaded region beneath the GMMs for the model in question.

The GMM logic tree adopted for FR2020 is the simplest of the three, using four ground motion models each assigned an equal
weight of 0.25 (Ameri, 2014[1]; Abrahamson et al., 2014; Cauzzi et al., 2015 [with variable reference $V_{S30}$]; Drouet and Cotton, 2015 [using rupture distance and with 10 MPa stress drop for large magnitude events]). Two of these models (Ameri, 2014; Drouet and Cotton, 2015) are based exclusively on French seismological data, while Abrahamson et al. (2014) is fit to records from the NGA West 2 dataset (global in scope but with most records originating from California), and Cauzzi et al. (2015) is fit predominantly to Japanese strong motion data (supplemented by some records from other regions of the globe). None of
the selected GMMs is based on the pan-European RESORCE ground motion data set (Akkar et al., 2014b), although Drouet et al. (2020) indicate that several of the GMMs that were derived using pan-European ground motion data were considered in the selection process. The analysis to support their model selection is based on the exploration of the model space of the GMMs using Sammon's maps (Scherbaum et al., 2010), which reveal that the four models are relatively well separated within the model space described by all pre-selected GMMs and by a set of reference models derived from the mean of the considered
GMMs scaled up and down (representing stress drop variation) and with faster or slower attenuation. In this sense, the multi-model logic tree accounts for epistemic uncertainty in both the model functional form as well as the geophysical properties of the target region, the latter being represented by the different GMM source regions implicit within the selected models: France (Ameri, 2014; Drouet and Cotton, 2015), Western United states (Abrahamson et al. 2014) and Japan (Cauzzi et al. 2015).

---

[1] The original paper of Drouet et al. (2020) indicated that Ameri et al. (2017) is adopted here; however, discussions with the authors revealed it was in fact the earlier Ameri (2014) model used.





The GMM logic tree for DE2016 is initially based on a multi-model approach, identifying five models identified as suitable for application to Germany (Akkar et al., 2014a; Bindi et al., 2014; Derras et al., 2014; Cauzzi et al., 2015; Bindi et al., 2017), but adds to each of these models a set of scaling factors to the median ground motions (0.7, 1.0, 1.25 and 1.5) to account for epistemic uncertainty in regional stress drop. Of the five models selected, Akkar et al. (2014a), Bindi et al. (2014) and Derras et al. (2014) are fit to data from the pan-European RESORCE data set, Cauzzi et al. (2015) fit to predominantly Japanese data

(as explained), and Bindi et al. (2017). The latter is fit to NGA West 2 data but using a simpler functional form than the NGA West 2 GMMs, which more suited for the level of parameterization commonly found in moderate to low seismicity regions. The DE2016 GMM logic tree combines both a standard multi-model approach with elements of a scaled backbone approach to capture some of the uncertainty in the underlying seismological properties of the target region; hence, we refer to it as a hybrid multi-model and backbone GMM logic tree.


Grünthal et al. (2018) outline several critical issues that influence their decision-making process: i) different strengths of the different databases of ground motion (e.g. tectonic similarity for Europe [RESORCE], wealth of short distance records [NGA West 2], detailed site parameterization [Japan – Cauzzi et al., 2014]), ii) variation in functional form and how this influences ground motion prediction for small-to-moderate magnitude events, and iii) the observation of several earthquakes with higher

than average stress drop in stable regions of France, Germany and the UK. The multi-model approach and the choice of models selected largely addresses the first two of these issues. Three different datasets (RESORCE, NGA West 2 and Japan) are represented, which also implicitly incorporate GMM source-region to source-region variability (i.e., Europe, Western US, Japan). The highest weight [0.5] assigned to the three GMMs derived from RESORCE and then split evenly between the three models therein, while the Cauzzi et al. (2015) and Bindi et al. (2017) models receive equal weights of 0.25. Functional form

variation and parameterization is accounted for by mixing classical random effects models (each with slight differences in functional form) with purely data-driven neural network models (Derras et al. 2014). In practice this approach is similar in outcome to that adopted in the FR2020 model, with the same three source regions represented (albeit the FR2020 selection uses France-specific models rather than European models) and with the "local" region receiving a weight of 0.5 and the other two a weight of 0.25 each. The two sets of GMMs for the DE2016 and FR2020 models are compared in Figures 6 and 7 in

terms of their range of median ground motions (Figure 6) and their aleatory uncertainty $\sigma_T$ (Figure 7).

The uncertainty stress drop is the motivation behind adding the additional scaling factors, which capture both the possibility that stress drop may be lower in Germany than in the respective source regions of the models (0.75) as well as the possibility that it is higher (1.25 and 1.5). Weights of 0.36 are assigned to each of the 1.0 and 1.25 scaling factors, while the outer branches

(for lower than average or much higher than average stress drop) are assigned smaller weights of 0.14 each. This pushes the balance of the weight toward higher stress drop in Germany.



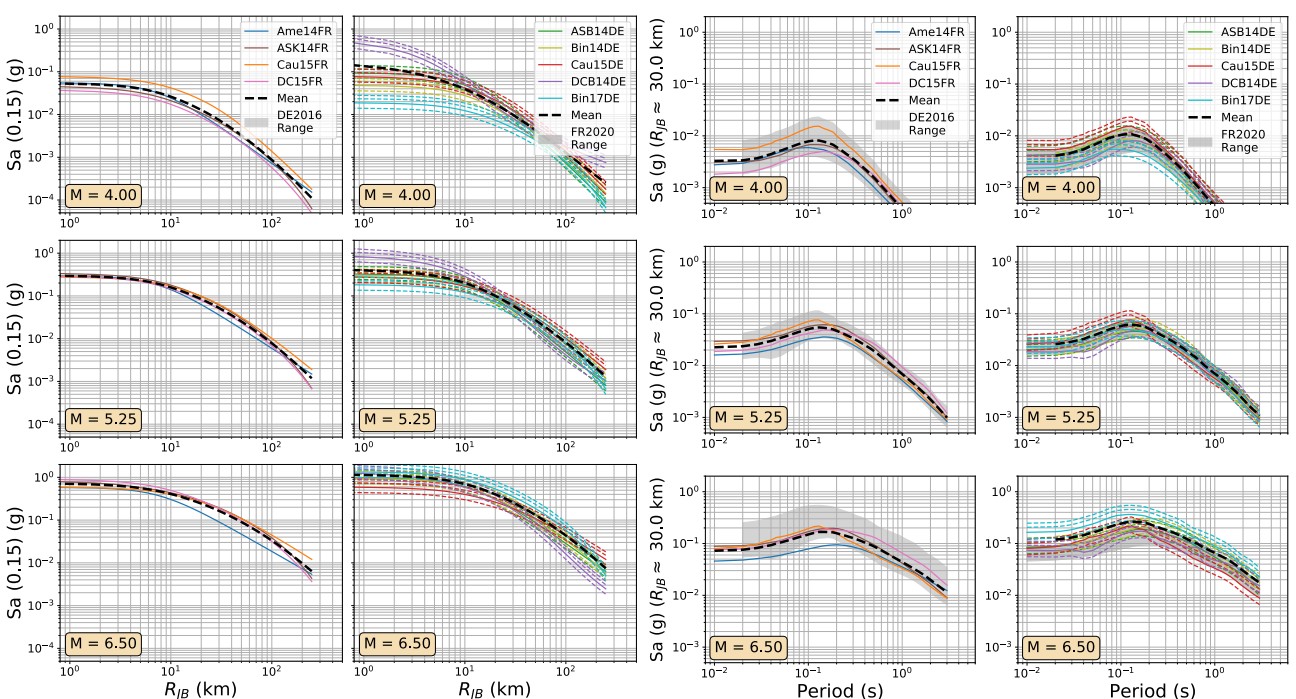

**Figure 6:** Trellis plots comparing the median ground motions of the GMM selections of the FR2020 and DE2016 logic trees. (left)
Attenuation with distance for Sa (0.15 s) for $M_W$ 4.0, 5.25 and 6.0, and (right) scaling with period at a site $R_{JB}$ 30 km from the source
for $M_W$ 4.0, 5.25 and 6.0. The range of values from the compared models is shown by the grey shaded region in each plot, while the
weighted median ground motion from the logic tree weights for each model is shown by the dashed black lines.

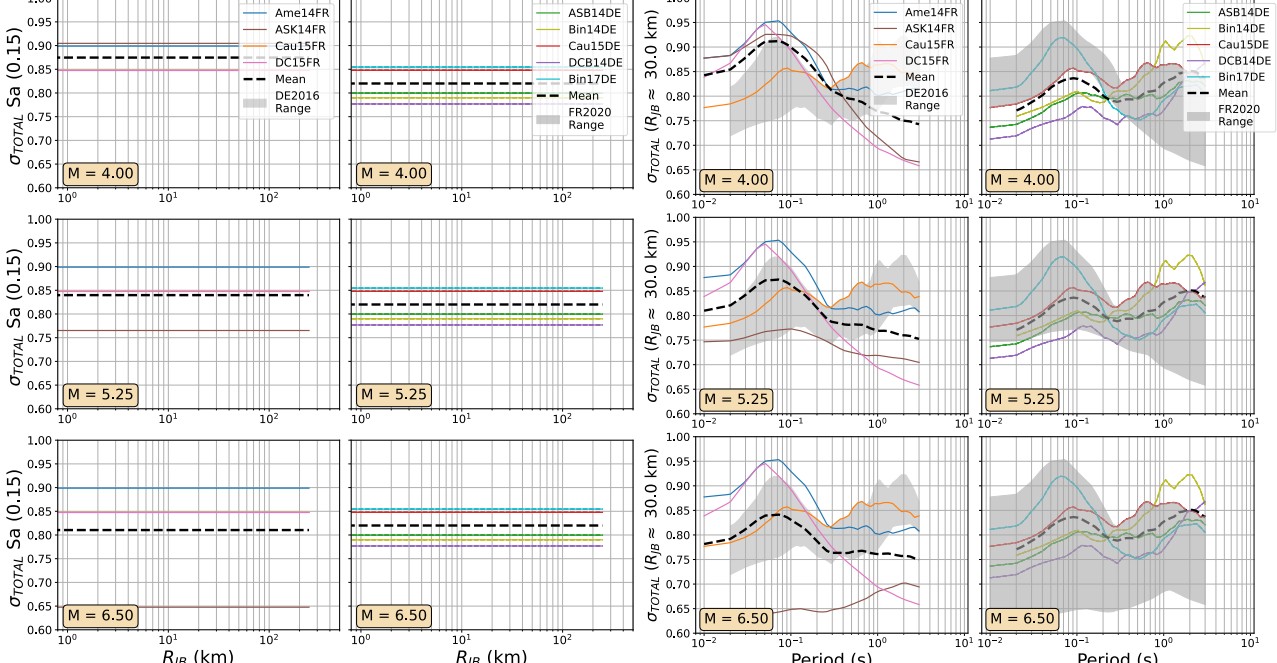

**Figure 7:** As Figure 6, comparing the aleatory uncertainty distributions of the FR2020 and DE2016 GMM logic trees






Compared to the strategies adopted for FR2020 and DE2016, the ESHM20 model has taken a different approach to defining a GMM logic tree that captures the centre, body and range of the technically defensible interpretations of available data, and it does so by abandoning entirely the multi-model concept in favour of a regionalized scaled backbone logic tree. The full explanation of the logic tree, including both its motivation and calibration, is given in Weatherill et al. (2020). This change in

approach is motivated in large part by the development of the Engineering Strong Motion (ESM) database and flatfile (Lanzano et al., 2019), which increases by nearly an order of magnitude the number of ground motion records available in Europe, particularly those of small-to-moderate magnitude earthquakes including many more from France and Switzerland than in RESORCE. The backbone GMM is fit to this data set (Kotha et al., 2020), but with such a large volume of data additional random effects are included to capture region-to-region variability in the stress parameter scaling of the model ($\delta L2L_l$) and in

the attenuation ($\delta c_3$ – where $c_3$ is the coefficient of the anelastic attenuation term of the model). These two random effects are both normally distributed variables with means of 0 and standard deviations of $\tau_{L2L}$ and $\tau_{c_3}$ respectively, and individually they quantify the total regional variability in stress parameter and residual attenuation within Europe. For regions with little or no ground motion data, the distributions of $\mathcal{N}(0, \tau_{L2L})$ and $\mathcal{N}(0, \tau_{c_3})$ are mapped into sets of discrete branches using the method of Miller and Rice (1983), making the model a scaled backbone model. Where data are available the distributions can be

adjusted to reflect the local stress parameter or attenuation properties implied by the data, thus the model is also regionalisable.

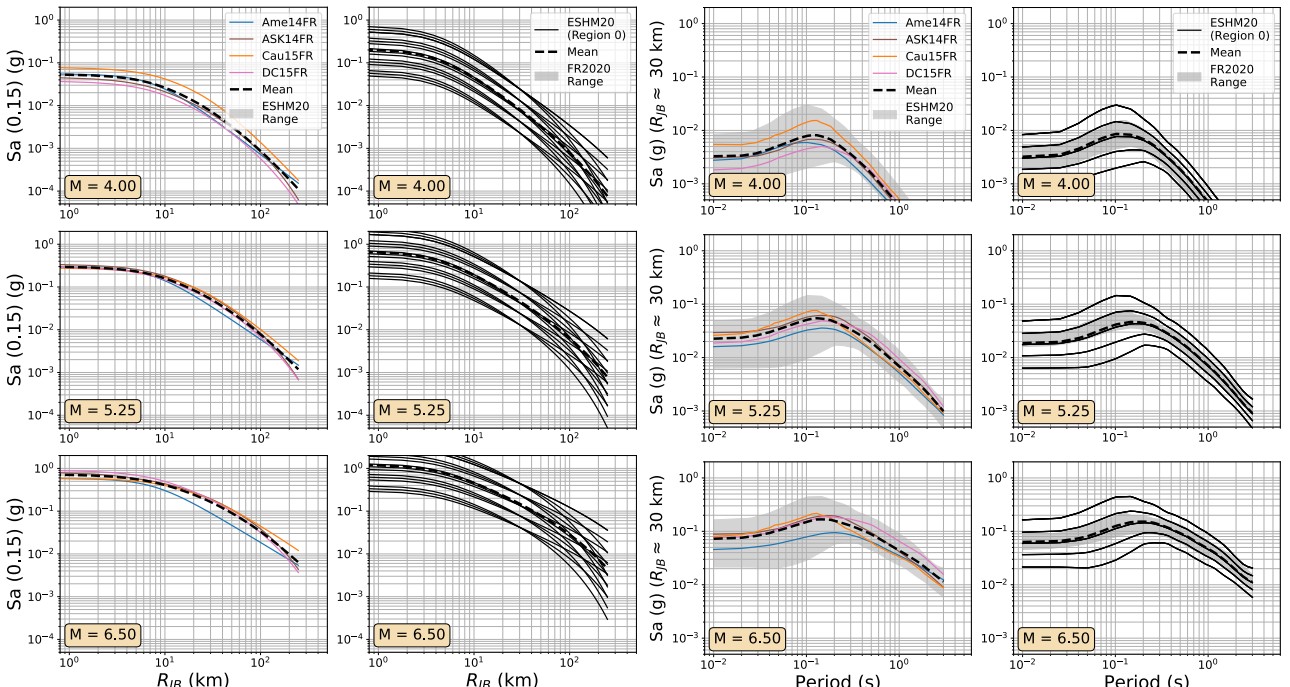

**Figure 8. As Figure 6, comparing the median ground motions of the ESHM20 and FR2020 GMM logic trees**

Even in the larger ESM flatfile there are few events from Germany, and those that are present are almost all located in the Upper Rhine Graben and Alpine Foreland. In France the majority of earthquake and records come from the Alpine and Pyrenees regions. Observations were available for the regions where $\delta c_3$ could be calibrated, so regions of similar $\delta c_3$ were grouped together to differentiate between regions of slower, average, or faster attenuation. These differences are reflected in the model, where the attenuation parameters backbone GMM for sites in these regions are adjusted to incorporate these

differences. Altogether, the regionalized scaled backbone logic tree maps the unadjusted (un-regionalised) $\delta L2L_l$ term into 5 branches and the regionalized $\delta c_3$ term into three branches, resulting in 15 GMM branches altogether. The median accelerations predicted by ESHM20 GMMs are compared against those of FR2020 and DE2016 in Figure 8 and 9 respectively, and the aleatory uncertainties in Figures 10 and 11.

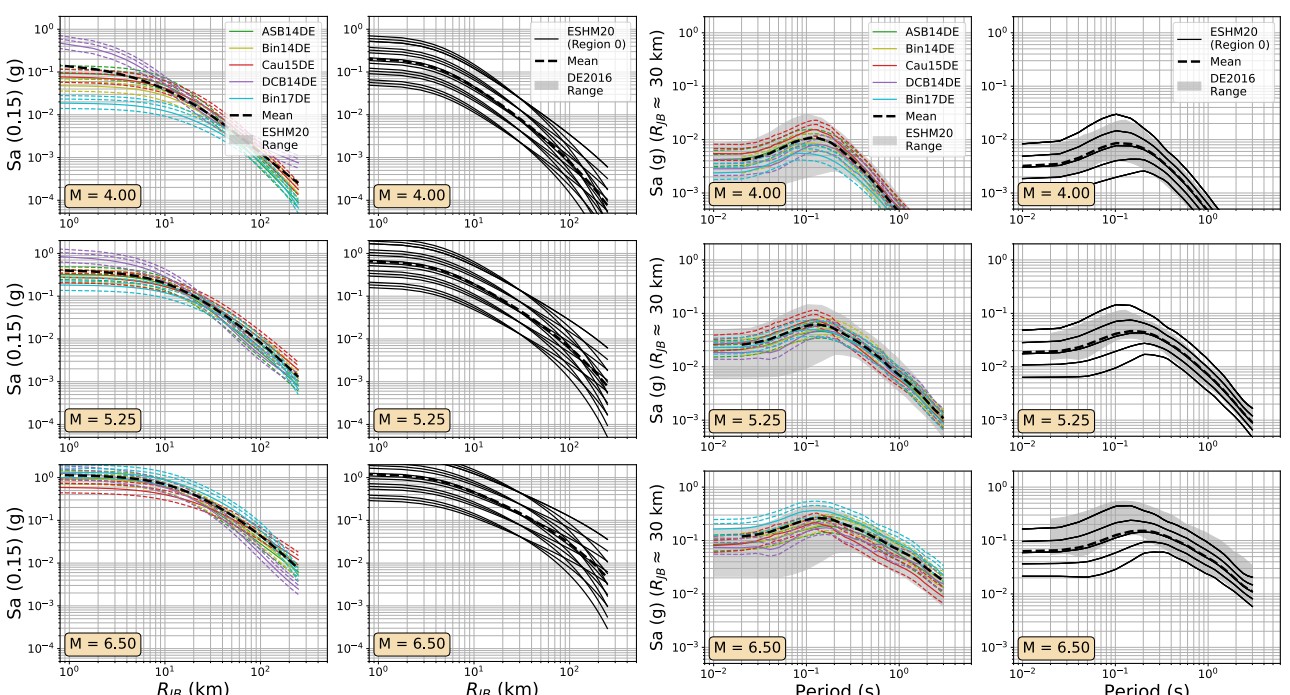

**Figure 9: As Figure 6, comparing the median ground motions ESHM20 and DE2016 GMM logic trees**

Among the most important trends to be seen in the plots in Figures 6 to 11 are the general tendencies toward higher median ground motions at short distances and small magnitudes for the GMM logic trees of the DE2016 and ESHM20 model compared

to that of FR2020. For larger magnitudes the trends reverse, and it is the ESHM20 GMM logic tree that provides a lower central tendency in the ground motions. At intermediate magnitudes and distances, where we are best constrained by data, ESHM20's GMM logic tree tends toward lower short period motions at most magnitudes and distances, while longer period motions are comparable. We note, however, the very high and very low stress parameter branches of the ESHM20 GMM logic




tree that envelope the range of values in the plots have very little weight associated to them, and it is the three more central

branches that have the greatest influence on the mean hazard.

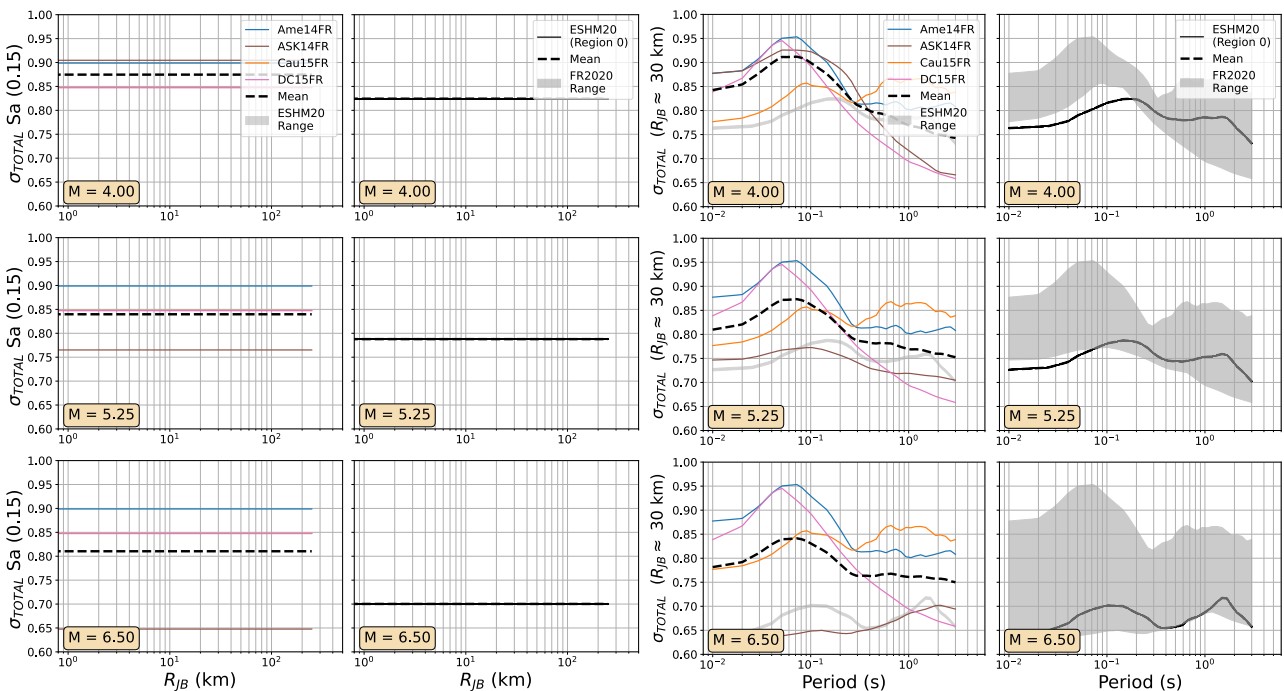

**Figure 10. As Figure 7, comparing the aleatory uncertainty distributions of the FR2020 and ESHM2020 GMM logic trees**

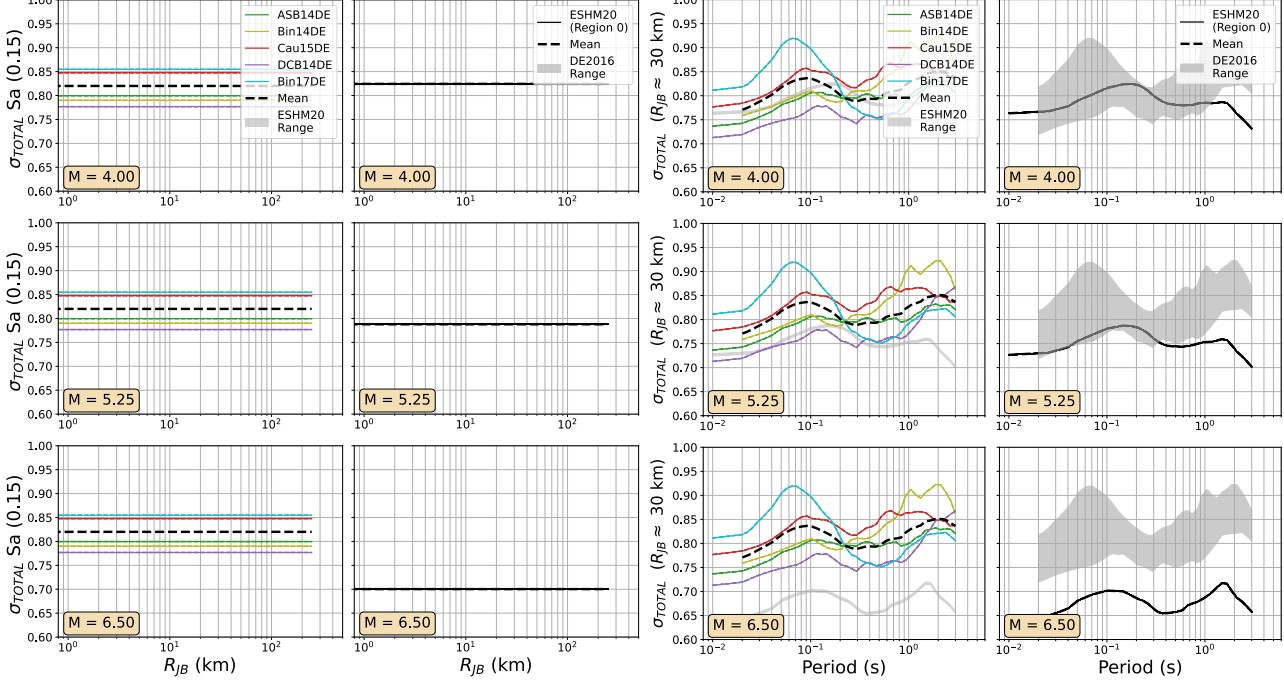

**Figure 11: As Figure 7, comparing the aleatory uncertainty distributions of the ESHM2020 and DE2016 GMM logic trees**




For the DE2016 and FR2020 comparisons, the DE2016 GMMs tend to skew higher. This reflects the influence of the stress drop scaling, where more weight is put on toward the scaling factors greater than or equal to 1.0. Without these adjustments the GMM selections would likely have returned a similar centre and range of ground motions, except at near source distances ($R_{JB} < 10$ km) where the Derras et al. (2014) GMM with the point-source to finite rupture distance correction seems to extrapolate toward much higher motion than the other models.

For the aleatory variability the ESHM20 is based on a scaled backbone model with no branches for epistemic uncertainty this parameter, so the range of $\sigma_T$ collapses to a simple line. What is evident, however, is the heteroskedastic variability that is present in the ESHM20 model and in the Abrahamson et al. (2014) model. The results in lower $\sigma_T$ at high magnitudes, which in turn lowers the aleatory uncertainty in the ESHM20 model compared to the other GMM logic trees and increases the range of $\sigma_T$ in the FR2020 model (albeit the range is being controlled by only one model). We also observed that the ESHM20 model shows a lower aleatory variability in general compared to the spread found in other GMM logic trees. Two factors play a role here, the first is that the Kotha et al. (2020) model was derived using robust linear mixed effects regression that down-weights outlier values, and the second is that the ESHM20 GMM implementation adopts different site-to-site variability ($\phi_{S2S}$) for the cases when the site condition ($V_{S30}$) is measured and when it is inferred from a proxy (Weatherill et al. 2022). For the measured $V_{S30}$ case, which is the one being considered in the ESHM20 application, $\phi_{S2S}$ is reduced compared to most other GMMs shown here because it is fit to the site-to-site variability of the subset of stations with measured $V_{S30}$, while most other models have calibrated this variability based on records from stations that mix measured and inferred $V_{S30}$.

**3. Harmonising Model Implementations into a Common Software Format**

We have so far looked at some of the fundamental differences in the seismic hazard inputs between the three national seismic hazard models, and though there are different approaches and philosophies underpinning each there are also many key similarities, most notably in the types of sources being adopted (i.e., uniform area zones, smoothed seismicity and, in the case of ESHM20 and DE2016, active fault surfaces). An important difference, however, is not just in the construction of the inputs but how they are processed in the PSHA calculation. Here the PSHA calculation software plays an important role. Each of the three models was implemented in a different PSHA software: FR2020 used a proprietary software developed by Fugro that is based on a customized version of the FRISK88 (McGuire, 1976); DE2016 also used a proprietary software that is their own customization of FRISK88 for the area and fault sources, which was combined with their own software code to implement smoothed seismicity PSHA; ESHM20 was developed using OpenQuake (Pagani et al., 2014).




Our first major objective in this work was to harmonise all three models into a common format around the OpenQuake-engine seismic hazard and risk software. This harmonization serves multiple purposes. The first is to migrate the models from the proprietary software in which they were originally implemented and to support them using and open-source software so that
they can be reproduced by other parties. The second purpose is the main objective of this paper, which is to define a common representation of hazard inputs and outputs that will allow for the quantitative comparisons shown in sections 4 and 5. Finally, OpenQuake includes both a seismic hazard and a seismic risk calculator, which in combination with the exposure and vulnerability models provided as part of ESRM20 allows us to explore implications of the different models in terms of seismic risk. This latter objective will, however, be the subject of a future work and is beyond the scope of the current paper.


**3.1 PSHA Software Comparisons: Rationale and Applications**

The fundamental framework for PSHA is largely unchanged changed since its establishment by Cornell (1968) and McGuire (1976), with arguably the most notable evolutions in practice emerging with the earthquake rupture forecast (ERF) approach (Field et al., 2003) and more widespread usage of Monte Carlo techniques (e.g., Ebel & Kafka, 1999; Musson, 2000; Weatherill
& Burton, 2010; Assatorians & Atkinson, 2014). These latter adaptations do not alter this core probabilistic framework but rather they evaluate it in a manner that may be flexible or better suited to incorporate new modelling developments or provide input into a broader range of applications. Yet despite the robustness of the conceptual probabilistic seismic hazard integral, different PSHA software can be remarkably divergent in the way the input source and ground motion models are processed and translated into the PSHA framework.


Differences between PSHA software can be broadly grouped into three categories:

*Irreconcilable discrepancies* owing to fundamental differences in software operation. These can include characterization of the seismic source and/or magnitude frequency relation and their discretisations within the hazard integral, treatment of rupture finiteness in distributed seismicity sources and its scaling with earthquake magnitude, calculation of fault rupture to site
distances, and evaluation and/or approximation of the statistical density functions to retrieve probabilities of exceedance of ground motion. Such differences can be identified but not necessarily replicated from one software to another without significant changes to the code.

*Implementation discrepancies*, which mainly refer to bugs or errors in the source codes themselves, potential instability due to
rounding errors, or different interpretations of ambiguously described features or parameters in implemented models such as GMMs. These can be identified and resolved by following quality assurance procedures, and greatly assisted by model authors providing open-source implementations of their models.

*Free modelling parameters and configuration choices that* allow users to control the operation of the software but that are
seldom fully documented (particularly in scientific papers). These may resemble more the irreconcilable discrepancies if one



software implements a part of the hazard calculation in a flexible manner that affords the user control of the operation, while another software may hard-code this same process and afford the user no control.

The way that different software characterise common elements of a PSHA calculation, and the corresponding impacts on the resulting hazard curves, have been evaluated as part of the PEER Probabilistic Seismic Hazard Code Verifications (PEER Tests hereafter) (Thomas et al., 2010; Hale et al., 2018). These are elemental PSHA calculations usually comprising a single source, ground motion model and a limited number of target sites with fixed properties, which are designed specifically to assess how the different software approach a particular modelling issue. The results are compared against either "exact" solutions calculated by hand, where possible, or against the range of curves determined from the participating PSHA codes when the problem cannot be evaluated by hand.

The PEER Tests have been particularly insightful in identify how and why PSHA codes diverge, especially given that many codes participated to them (both proprietary and open source) that are widely used in commercial application. As they are elemental in nature, however, they cannot necessarily predict the extent to which different codes will yield different outputs for seismic hazard at a given location, where many modelling differences come into play. The importance of this type of application and the benefits of multi-software implementations of a seismic hazard model as part of a quality assurance (QA) process for the design of critical facilities has been highlighted emphasized by Bommer et al. (2015) and Tromans et al. (2019), among others, and is becoming more widely used in practice. The QA application is only one context, however, and arguably a favourable one in which multiple parties are involved and resources often made available to document and debate the implementations, and to resolve discrepancies as and when they emerge.

A more relevant for the case at hand is migration of an existing or established hazard model from one software to another. Here the challenges are different, as the existing model forms the reference, and the new software may need to replicate the behaviour of the previous one in order to ensure consistency in the outputs. In some cases, if the new software user is different from, or does not have support of, the original software developer and the source code of the software is closed, then there can often be critical elements of the PSHA calculation process to which the user is themselves blind. In this instance complete agreement between the existing and migrated models may not be possible due primarily to the irreconcilable differences between software highlighted above. Instead, a target level of "acceptable agreement" between previous and new implementation needs to be defined (e.g., Allen et al., 2020; Abbot et al., 2020).

In the migration processes described in this section we set a target level of agreement in terms of the OpenQuake calculated seismic hazard curves at given target sites agreeing with those produced from the original PSHA software code agreeing to within ± 10 % *annual probability of exceedance (APOE)* for the corresponding range of ground motion *intensity measure levels (IMLs)* for APOEs greater than $10^{-4}$ (corresponding to a return period of approximately 10,000 years). Though in many



cases agreement can be achieved for lower APOEs, the irreconcilable differences due to issues of discretization, rounding, numerical instability etc. may begin to influence the extreme tails of the distributions that assume greater importance at these longer return periods. An APOE of $10^{-4}$ is sufficient to span the range of return periods considered for conventional design building codes, which reflect the applications for which these specific hazard models are intended. As both the FR2020 and DE2016 models have logic trees we undertake comparisons in two steps, the first comparing specific branches of the logic tree

to ensure broad agreement over source and ground motion model combination, the second comparing the curves in terms of the respect means and quantiles. We note that from the seismic hazard curves similar agreement targets could be set in terms of the IMLs for a fixed range of APOEs, which may be slightly more intuitive. Both options were explored, and no cases were found in which the agreement in curves for the IMLs failed to reach the set ± 10 % target when the agreement in terms of APoEs did. As all three software considered return seismic hazard curves in terms of PoE for a user-input set of IMLs, and

statistics of means and quantiles were calculated based on PoE, we opted to use APoE as the variable for the comparisons to avoid introducing potential discrepancies from different interpolation approaches. Summaries of the migration issues for both FR2020 and DE2016 can be count in Electronic Appendix A Notes 3, 4 and 5, with further details of the issues encountered in the migration of FR2020 to OpenQuake can be found in Weatherill et al. (2022). Illustrative comparison plots of the two software implementations both for national seismic hazard maps and seismic hazard curves at selected locations can be seen

in Electronic Appendix B.

### 3.2 Defining Means and Quantiles

In OpenQuake the mean is calculated as the weighted *arithmetic* mean of the probabilities of exceedance (PoE) for each given intensity measure level (IML). Similarly, quantiles are determined based on the probabilities of exceedance for each intensity

measure level by sorting the PoEs from lowest to highest at each IML and interpolating the corresponding cumulative density function to the desired quantile values (typically 0.05, 0.16, 0.5 [median], 0.84 and 0.95). As OpenQuake adopts the earthquake rupture forecast (ERF) formulation for the PSHA calculation (Field et al., 2003), all hazard statistics are extracted from the probabilities of exceedance rather than the rates of exceedance. This formulation of the mean and quantiles represents one of several different ways of retrieving this term. Other PSHA software may apply the statistics to the IMLs for a given PoE and/or

work with the geometric rather than arithmetic means and each approach yields different results. From communication with the model developers, we verified that FR2020 defines the mean hazard as the arithmetic mean of the probabilities of exceedance, while for DE2016 the means are based on the arithmetic mean of the annual rates of exceedance. For consistency with OpenQuake, in the comparisons of means and quantiles show we have retrieved these values from the complete suite of hazard curves and processed them identically, rather than taking the mean or quantiles from the software itself.


### 3.3 Source-to-Source Correlation in MFR Epistemic Uncertainties

We have seen in section 2 how the three different models attempt to translate the uncertainty on $a$, $b$ and $COV(a, b)$ into the logic tree, and how this yields quite different distributions of activity rates. An issue that is not discussed is the issue of source-


to-source correlation in the MFRs. To summarise, consider an idealized model with just four area sources, each with their own truncated Gutenberg-Richter MFR, and a corresponding logic tree with three branches for uncertainty on $a$ and $b$ (e.g., $-\Delta \cdot (a,b), \overline{(a,b)}, +\Delta \cdot (a,b)$)and three for uncertainty on $M_{MAX}$ (e.g., $M_{MAX}^{LOW}, \overline{M_{MAX}} + M_{MAX}^{HIGH}$). If the MFRs are fit independently for each zone then the resulting logic tree would need to permute every combination of the MFR parameters for each source, which would in this simple case results in $9^4 = 6561$ logic tree end branches, i.e., $(N_{BRANCHES})^{N_{SRCS}}$. Applying this same logic to the area source zonations for DE2016, for example, we have between 31 and 107 sources per model and 20 MFR branches, which would result in between $20^{31}$ to $20^{107}$ logic tree branches *for each source model*. This is clearly intractable for any PSHA calculation software and OpenQuake cannot even construct such a logic tree from which to sample. A common alternative is to assume perfect correlation between the sources, which in the idealized case would be to apply the same branches (e.g. $-(a,b)\big|M_{MAX}^{low}, \overline{(a,b)}\big|M_{MAX}^{LOW}, +(a,b)\big|M_{MAX}^{LOW}, -(a,b)\big|\overline{M_{MAX}}, \dots, +(a,b)\big|M_{MAX}^{HiGH}$) to all of the sources at the same time. This results in a more manageable logic tree of just 9 branches in the simple idealized case and 20 MFR branches per source model in the DE2016 case.

Both DE2016 and ESHM20 adopt discrete MFRs for each of the sources meaning that in order to execute the calculation perfect correlation between sources had to be assumed in both cases. By sampling the MFRs for each source separately in the 100 branches, however, FR2020 is preserving independence in the source model MFRs. This issue of correlation can impact on the outcomes of the hazard as the assumption of perfect source-to-source correlation in MFRs could conceivably assign disproportionately large weights to the extreme cases that all sources may have higher or lower activity rates. This inflates the uncertainty meaning that the resulting hazard distributions may be larger than intended and potentially skewing the mean toward higher values compared to the case in which MFR epistemic uncertainties are characterized independently for each source. Work is currently ongoing to explore this issue in further detail and its impacts on seismic risk assessment for a country.

### 3.4 Calculation Scale

A final issue of PSHA implementation relates to the scale of calculation, by which we refer to the volume of data and, by extension, the CPU time and RAM needed execute the PSHA for logic trees of this size. Each of the three software address this differently, and as two of the software are proprietary we have not been able to benchmark the calculations. For OpenQuake, however, this type of logic tree with many source- and MFR-branches is not efficiently handled *at the time of writing*. The main reason for this is that for each source model and MFR branch a new earthquake rupture forecast is constructed. This requires re-calculation of distances and ground motions for each logic tree branch MFR branch, increasing both the CPU and RAM requirements. Calculations here were run on a 192 CPU server with 760 Gb RAM, and this was insufficient to execute the calculations in a single run. Instead, the models for FR2020 and DE2016 were split into subsets of branches and the resulting hazard curves later recombined and post-processed to retrieve the mean and quantiles. It is hoped





that future efforts will be undertaken to improve the efficiency of the calculations for this type of epistemic uncertainty, which
is commonly applied in regions of low to moderate seismicity.

## 4. Quantitative Comparisons of the Seismogenic Source Models by Visualising Activity Rate Model Space

In section 2 we showed the overall structure of the different models, contrasting some of the assumptions behind them and
looking in detail at the France-Germany border region to understand the differences in catalogues, definitions of source models,
and the fitting and characterization of the recurrence models. Though this process brings to light some of the main factors that
will go toward explaining the differences in seismic hazard results shown in the next section, it is also important to be able to
quantify and interpret differences in the two primary components of the PSHA model: the seismic source model and the ground
motion model. Comparisons at this point can be particularly useful as they can allow us to understand the cumulative impact
of the diverging steps that have led to the construction of the respective source and ground motion models before these are
then integrated into the PSHA calculation. A crucial motivation for the migration of the PSHA models into a common software,
as described in detail in section 3, is to have the three models represented in a common format that allows us to isolate the
model-to-model differences from the software differences. In this section all the analysis is working with the OpenQuake
implementations of the models rather than the original implementations (in the case of FR2020 and DE2016).

### 4.1 Interpreting the Seismogenic Source Model Space using Descriptive Statistics

Section 2 explained how all three models share some similarities in the source types that they are using, but their differences
too. As each model is adopting a logic tree with epistemic uncertainty on both the source types and recurrences, how can one
quantitively compare not just the sources but their respective distributions? The starting point is to render each source into a
common representation allows for quantitative comparisons of the models and their respective distributions. Each source
branch of each model is translated into a three dimensional array $\lambda(\phi, \theta, M)$ of latitude, longitude and magnitude, with each
cell containing the incremental rate of activity for each the corresponding longitude, latitude and magnitude bin $\lambda_{ijm}$, where
$i = 1, 2, \dots, N_\phi$ corresponds to the longitude bin, $j = 1, 2, \dots, N_\theta$ to the latitude bin, and $m = 1, 2, \dots, N_m$ to the magnitude bin.
For area sources the rate of the whole source is partitioned into each grid cell according to the proportion of total area in each
cell, for gridded seismicity the rate of all sources in which the centroid of the source grid cell falls, while finally for the fault
sources the seismicity rate per cell is partitioned according to the proportion of the fault's surface projection that intersects the
cell. All seismogenic sources here are shallow crustal sources, so although hypocentral depth is relevant to the seismic hazard,
for the current purposes rates are not distributed across different depth layers.

Each source model logic tree branch $k$ of $N_k$ total branches is rendered into the 3D rate grid $\boldsymbol{\lambda_k(\phi, \theta, M)}$ and each grid is
associated with its respective logic tree branch weight. This relatively simple translation of the respective source models into
a common grid representation facilitates quantitative comparisons by virtue of simple descriptive statistics. For example,



Figure 12 shows the spatial variation in mean cumulative rate of seismicity above $M$ 4.5 for each of the three models, which is weighted by the logic tree branch weight for each source branch:

$$\overline{\lambda(\boldsymbol{\phi}, \boldsymbol{\theta}|M \geq 4.5)} = \sum_{k=1}^{N_k} w_k \cdot \sum_{m=1}^{N_M} \lambda_k(\boldsymbol{\phi}, \boldsymbol{\theta}, M_m) \cdot H[M_m \geq 4.5] \tag{5}$$

where $H[\cdot]$ is the Heaviside step function. Similarly, weighted percentiles can be extracted for each spatial bin, which we show in Figure 12 as the 16th and 84th percentiles. The minimum magnitude $m_{min} = M$ 4.5 is used in these comparisons as this is the common minimum magnitude in the PSHA calculations for all three models. Other values of $m_{min}$ could be compared depending on the relevant context; however, $m_{min} = 4.5$ is sufficient to illustrate the application here. From these descriptive statistics we can extract a measure of the centre and body of the activity rate distributions, the latter being illustrated in terms

of the weighted interquartile range in Figure 13. Note that the striations in the maps for the FR2020 model emerge from the gridded seismicity branches being regularly Cartesian spaced every 10 km, while the reference grid is in a geodetic system (longitude and latitude).

It is not our intention to provide a complete interpretation of all the features visible in these maps, though for the comparisons

of hazard in the France-Germany border region noteworthy differences include the relative activity of Albstadt Shear Zone (SE Germany) and the Upper and Lower Rhine Graben. The Albstadt Shear Zone is a particularly complex feature where the smoothed seismicity driven branches of the DE2016 and ESHM20 produce very localized zone of high activity while several area zonations (particularly those based on regional tectonics) do not isolate this region from the larger-regional seismicity. So higher quantiles tend to reflect the smoothed seismicity branches in which the ASZ is highly visible and lower quantiles reflect

the larger scale zonations where the ASZ is not present.




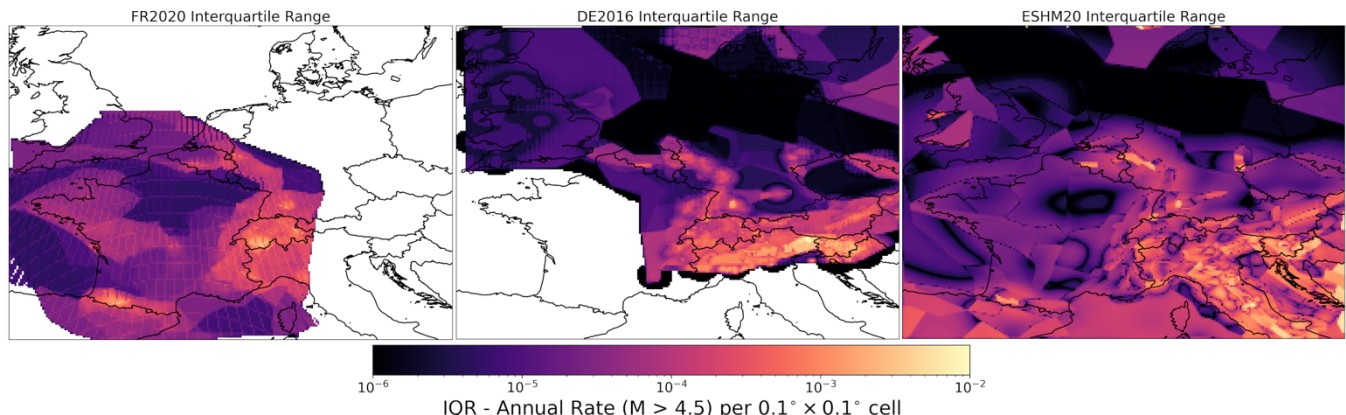

**Figure 12: Grids of activity rate for M ≥ 4.5 for FR2020 (top row), DE2016 (middle row) and ESHM20 (bottom row) in terms of mean rate (left column), 16th percentile (middle column) and 84th percentile (right column)**


**Figure 13: Interquartile ranges of activity rates from each source model logic tree: FR2020 (left), DE2016 (middle) and ESHM20 (right)**



Relative differences between the models can be quantified from this same characterization via the use of difference maps, both

for the mean activity rates (Figure 14, top) or for relative differences in the model range shown by the ratio of the interquartile

ranges (Figure 14, bottom). The difference maps present a somewhat incoherent picture, which is not unexpected given the

complexities and variations in the constituent source models.

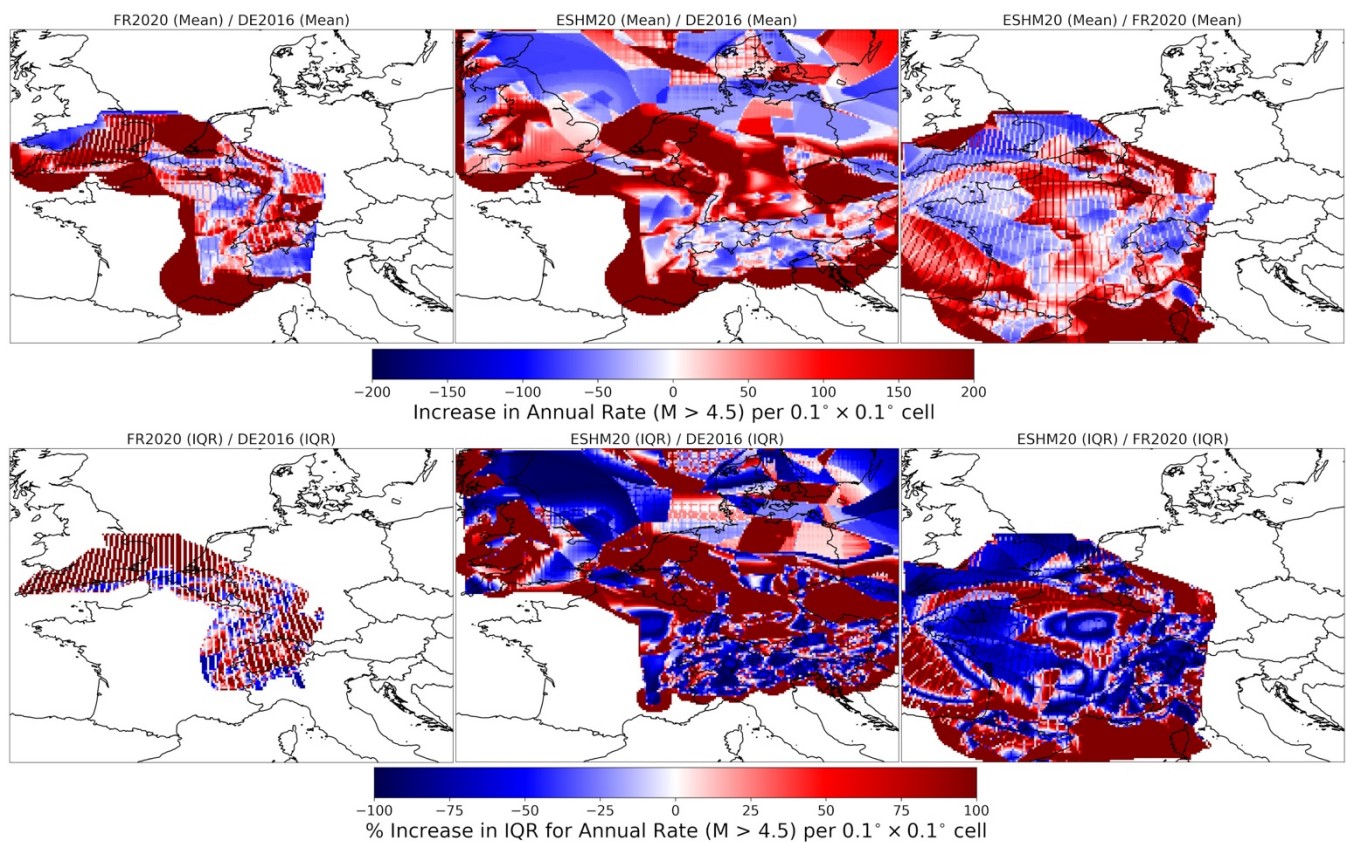

**Figure 14: Relative increase (in %) between the mean activity rate grids for each model comparison (top row) and the increase in interquartile range (%) (bottom row): FR2020 / DE2016 (left), ESHM20 / DE2016 (middle) and ESHM20 / DR2020 (right)**

**4.2 A Non-Parametric Statistical Approach**

Comparisons of the mean and quantiles of the rate distributions such as those shown in Figures 12 and 14 are certainly

important as they highlight regions where the underlying source models have a general tendency toward increased or reduced

activity. However, these metrics alone don't necessarily provide insight into the complete similarity of dissimilarity of the full

distributions of activity rates diverse, or how this divergence varies geographically. To visualize that sort of information we

can instead adopt metrics from information theory to help quantify dissimilarity between distributions: weighted Kolmogorov-

Smirnov Statistic ($D_{KS}$) and Wasserstein Distance ($D_{WS}$). If $\lambda_k(\phi, \theta, m_{min})$ is the rate grid for source branch $k$, with weight

$w_k$ then we can define for each complete source model logic tree a probability distribution $f_{MODEL}(\lambda_k|\phi, \theta, m_{min})$ at each




location, where $\lambda_k|\phi, \theta, m_{min}$ is the total activity rate in the spatial domain $(\phi, \theta)$ greater than or equal to a specified minimum

magnitude $m_{min}$.In the simplest case the spatial domain refers to each grid cell; however, this same process applies to any

spatial subdomain of the region enclosed by the original rate grid and could be applied to larger regions or somehow coarsened

with respect to the grid. If $f_{MODEL\_A}(\lambda_k|\phi, \theta, m_{min})$ and $g_{MODEL\_B}(\lambda_k|\phi, \theta, m_{min})$ are the respective empirical probability

density functions for the two full seismic source models $A$ and $B$ implied by their logic trees, then:

$$D_{KS} = \sup_{\lambda_k}|F_{MODEL_A}(\lambda_k|\phi, \theta, m_{min}) - G_{MODEL_B}(\lambda_k|\phi, \theta, m_{min})|$$ (6)

and

$$D_{WS} = \int_{-\infty}^{\infty}|F_{MODEL_A}(\lambda_k|\phi, \theta, m_{min}) - G_{MODEL_B}(\lambda_k|\phi, \theta, m_{min})| \, d\lambda_k$$ (7)

where $F_{MODEL_A}$ and $G_{MODEL_B}$ are the respective empirical cumulative density functions of models $A$ and $B$. The conceptual

definitions of these terms are illustrated in Figure 15, where we can see $D_{KS}$ as maximum absolute distance between the

empirical CDFs and $D_{WS}$ as total area enclosed between the CDFs. $D_{KS}$ is constrained to the domain $[0, 1]$, with 0 indicating

perfect agreement in the CDFs and 1 indicating no overlap in the respective ranges of $\lambda_k$, while $D_{WS}$ is constrained only by a

lower bound of 0 (total agreement). By working on the cumulative density functions, both terms account for the distribution

of weights in each of the logic trees.

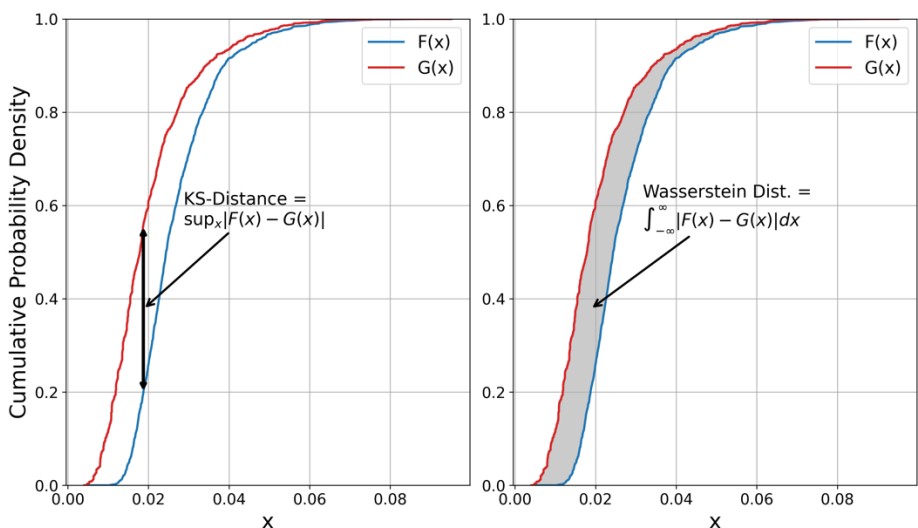


**Figure 15. Definition of the Kolmogorov-Smirnov (KS) statistic (left) and Wasserstein Distance (right) with respect to empirical CDFs $F(x)$ and $G(x)$**

With $D_{KS}$ and $D_{WS}$ we have metrics that allow us to assess spatial variation in the similarity between the effective rate

distributions predicted by two different models, which is shown for the combinations of FR2020/DE2016, ESHM20/DE2016

and ESHM20/FR2020 in Figure 16. The most immediate contrast between the maps using the two different metrics is the

apparent "noisiness" of the $D_{KS}$ metric compared to that of $D_{WS}$. The Kolmogorov-Smirnov statistic can appear to change




significantly over short distances, often highlighting boundaries of source zones, while Wasserstein Distances show a smoother transition, particularly in regions of higher seismicity. The sharp spatial contrasts and variability appear to be particularly

exacerbated for comparisons involving ESHM20. This behaviour may be anticipated by the conceptual definitions of the metrics shown in Figure 15, in which the largest absolute distance between empirical CDFs can change significantly even with relatively small changes in the shape of the CDF. In the empirical CDFs for $F_{MODEL_A}$ and $G_{MODEL_B}$, notable changes in shape may appear from one source zone to another due to changes in the MFRs for each of the zones, while in the case of ESHM20 the comparatively few MFR branches results in empirical CDFs that are more step-like, which results from having gaps in the

PDF that can arise due to coarse discretisation of the continuous distributions and/or transitions from one type of source or recurrence model to another. In this respect, $D_{WS}$ appears to be a better suited metric for interpretation as it is less sensitive to small changes in the empirical CDFs than $D_{KS}$. Focusing on this metric, in the France-Germany border regions we can see more coherent trends, such as greater divergence in the lower Rhine Graben than along the upper Rhine for all models, with the ESHM20 providing a notably divergent distribution here. Similarly, the Albstadt Shear Zone emerges as a point of

divergence between FR2020 and the other two models.

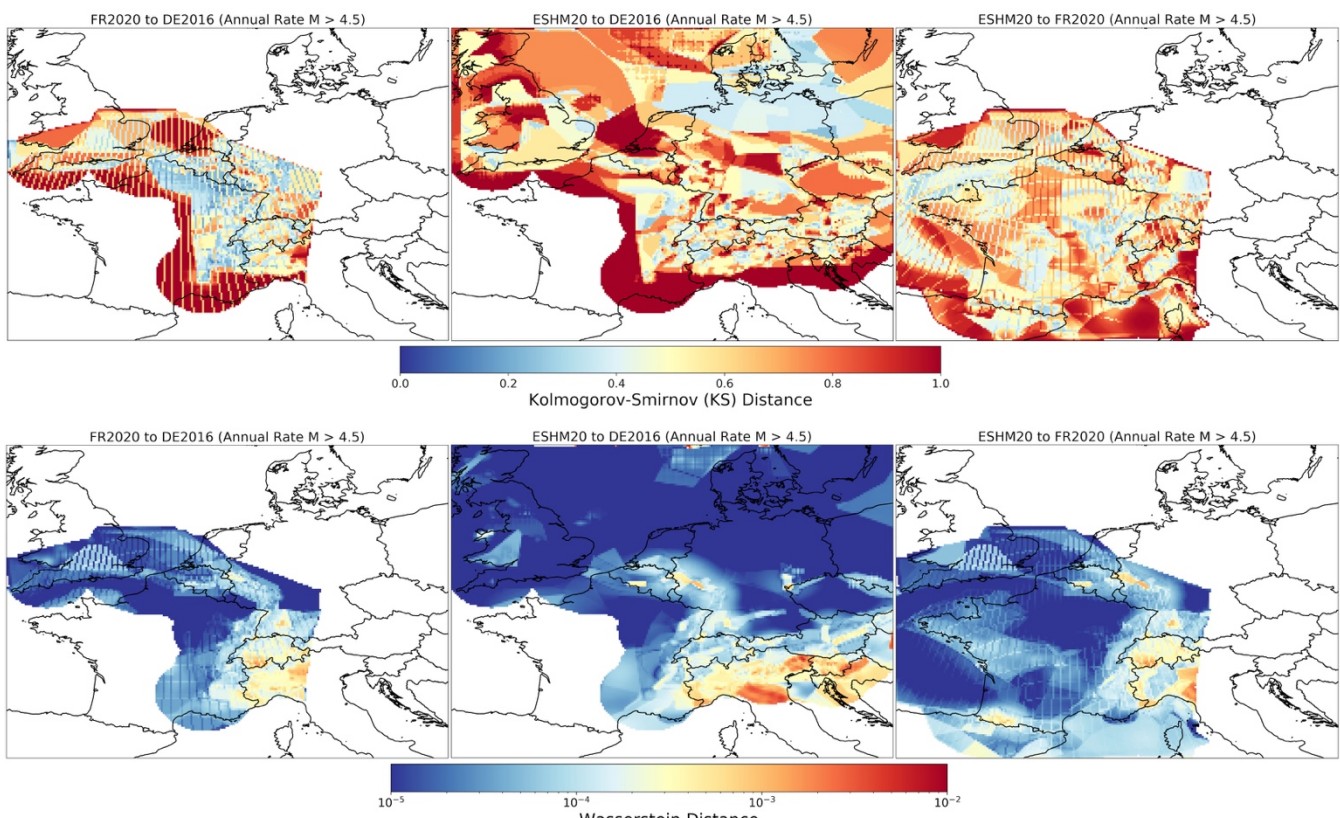

**Figure 16: Seismicity rate distribution differences between the models given in terms of KS Distance, $D_{KS}$ (top row), and Wasserstein Distance, $D_{WS}$ (bottom row): FR2020 to DE2016 (left), ESHM20 to DE2016 (middle), and ESHM20 to FR2020 (right)**






The approach of rendering each model into regular rate grids does allow us to make comparisons of the source models in a common framework and to interpret differences using simple descriptive statistics as well as through more information theoretic metrics. We contrast here the source models from the three different PSHA models (FR2020, DE2016 and ESHM20), though similar comparisons could be undertaken for successive generations of models, albeit one does not need to go back 795 more than one or two generations of regional scale model before concepts such as the logic tree are no longer found. From the comparisons of the source models shown here, a recommendation would be to compare models firstly via difference maps of mean rates, and potentially a small number of selected quantiles, then to apply $D_{WS}$ to be able to interpret quantitatively how and where the distributions diverge.

## 5. Quantitative Comparisons of the Seismic Hazard Model Results

With the components of the seismic hazard models compared in the previous section the obvious endpoint to this analysis is to undertake a comparison of the distribution of the seismic hazard results. To make such comparisons we limit the area of investigation to the France-Germany border region, stretching from the border with Switzerland in the south through to the Luxembourg border. The focus is now limited to this region as it is only here that we have sufficient overlap between all three models to capture contributions from sources up to the stated integration distance of 200 km. Though the Lower Rhine Graben 805 to the north is also of critical importance for understanding seismic hazard in Germany, this region is located at the very eastern extreme of the source models for France, thus the FR2020 sources do not provide complete coverage. Seismic hazard calculations have been run using the OpenQuake-engine implementations of each model for a target region enclosed by 5˚E – 9.5˚E and 47˚N to 50.5˚N, with target locations spaced every 0.05˚ ($\approx$ 3.5 – 3.7 km spacing EW and $\approx$ 5.5 km spacing NW). Mean hazard and its respective quantiles are calculated using the arithmetic mean of the probabilities of exceedances, rather 810 than the levels of ground motion. Hazard curves are determined for all models PGA and spectral accelerations at periods between 0.05 s and 3.0 s. Seismic hazard maps and corresponding difference maps for the 10 % probability of exceedance in 50 years are shown for three intensity measures (PGA, Sa (0.2 s) and Sa (1.0 s)) in Figure 17.

**Figure 17: (left) Probabilistic seismic hazard maps covering the France/Germany border region for PGA (top row), Sa (0.2 s) (middle row) and Sa (1.0 s) (bottom row) for 10 % PoE in 50 years. (right) Corresponding difference maps for the hazard comparing FR2020 / DE2016 (right column), ESHM20 / DE2016 (middle column) and ESHM20 / FR2020 (right column)**

As we had seen for the distributions of activity rate, comparisons of the resulting hazard maps for means and quantiles only reflect part of a larger picture. Instead, we can also frame the concept of similarity in hazard at a given probability of exceedance in terms of similarity or dissimilarity in the full distribution of hazard values emerging from the logic tree. Once again, we can invoke the two distances ($D_{KL}$ and $D_{WS}$) as measures of dissimilarity for a given acceleration level A with a P % probability in T years. In addition, we consider a third metric developed by Sum Mak (*personal communication)*, which we refer to as Overlap Index (*OI*). The *OI* is illustrated conceptually in Figure 18 for the distribution of ground motions from the FR2020 and DE2016. The distribution of hazard (here as PGA with a 10 % probability of exceedance in 50 years) is rendered into a histogram, with the weights of each value corresponding to its branch weight from the logic tree. *OI* between the distributions of seismic hazard from two different PSHA models at a given probability of exceedances is calculated from:





$$OI = \int_x \min\big(f(x), g(x)\big) \, dx \tag{8}$$

where $f(x)$ and $g(x)$ correspond to the observed probabilities densities of ground motion values for the two models

830 respectively. As with $D_{KS}$, $OI$ is bounded in the region $[0, 1]$ but here 0 indicates no region of overlap between the models and

1 a perfect agreement.

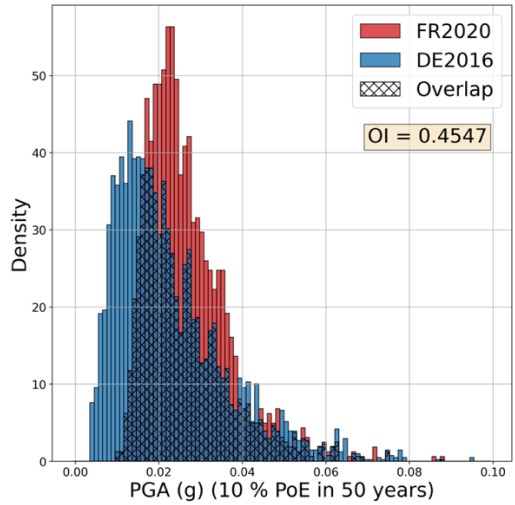

**Figure 18: Illustration of the Overlap Index (OI) between the distribution of hazard at a site using the FR2020 and DE2016 models**

The spatial distribution of dissimilarity between the full hazard models (in terms of the 10 % probability of exceedance in 50

years) can be mapped using the three metrics ($D_{KL}$, $D_{WS}$ and $OI$) shown in Figures 19 and 20 respectively. The difference

maps reveal several interesting features about the differences in the models in this region. Along the main channel of the Rhine

as it forms the border between France and Germany from Basel to near Karlsruhe, both the $D_{KS}$ and $D_{WS}$ measured indicate

less dissimilarity between the ESHM20 and DE2016 models, while for these same two models the $OI$ finds less overlap along

much of the entire Rhine Graben. Differences between the FR2020 and other models are clearly period-dependent in this same

region, with the Upper Rhine Graben seemingly in good agreement with other models for PGA and Sa (1.0 s). Yet for Sa (0.2

s) this same region is clearly illuminated as an area of significant disagreement. Dissimilarity seems to be lower in the northwest

of the target region close to the France-Luxembourg border, while it is in most cases at its greatest in northern Switzerland.

The Albstadt shear zone in the east is once again clearly highlighted, with the divergence between the FR2020 and other

models clearly visible.

The hazard maps and the corresponding dissimilarity maps show how the distributions of seismic hazard for a given IMT and

return period change with space, but these should also be complemented with more in-depth comparisons of the hazard curves

and uniform hazard spectra at specific locations. In Figure 21 we show two such comparisons for the cities of Saarbrücken, in

an area of lower hazard, and Strasbourg, which sits in the region of higher hazard along the Upper Rhine Graben. In addition,

these comparisons are shown in Appendix B alongside those of other towns close to the France-Germany border (Luxembourg,

Karlsruhe, Freiburg im Breisgau, and Basel). Here the full seismic hazard curves including the mean, 16[th] and 84[th] percentile are shown for Sa (0.15 s) (a period close to the peak of the UHS), alongside corresponding UHS for a 10 % and 2 % probability of being exceeded in 50 years. Saarbrücken sits in a region that we infer from Figures 19 and 20 shares a similar seismic hazard

distribution in the FR2020 and ESHM20 models but is notably lower in DE2016, while Strasbourg lies about halfway along the Upper Rhine Graben, a region where all three models seem to agree with one another. If we recall the comparison of the URG source zone in Section 2 (Figure 5) and the differences between the recurrence models for the three PSHA models found therein, the degree of agreement between the three models for Strasbourg is somewhat surprising. For both return periods the mean curves and UHS predicted by each model is falling within the 16[th] to 84[th] percentile of each of the others. Though this is

illustrative of the considerable range of ground motion values described by the 16[th] to 84[th] percentile, it does suggest a degree of consistency between them that may not be understood if one were to consider solely the changes in mean hazard.

**Figure 19: Spatial variation in the dissimilarity between distributions of seismic hazard values for a 10 % PoE in 50 years for PGA (top row), Sa (0.2 s) (middle row) and Sa (1.0 s) (bottom row) using KS Distance (left column set) and**
**Wasserstein Distance (right column set)**

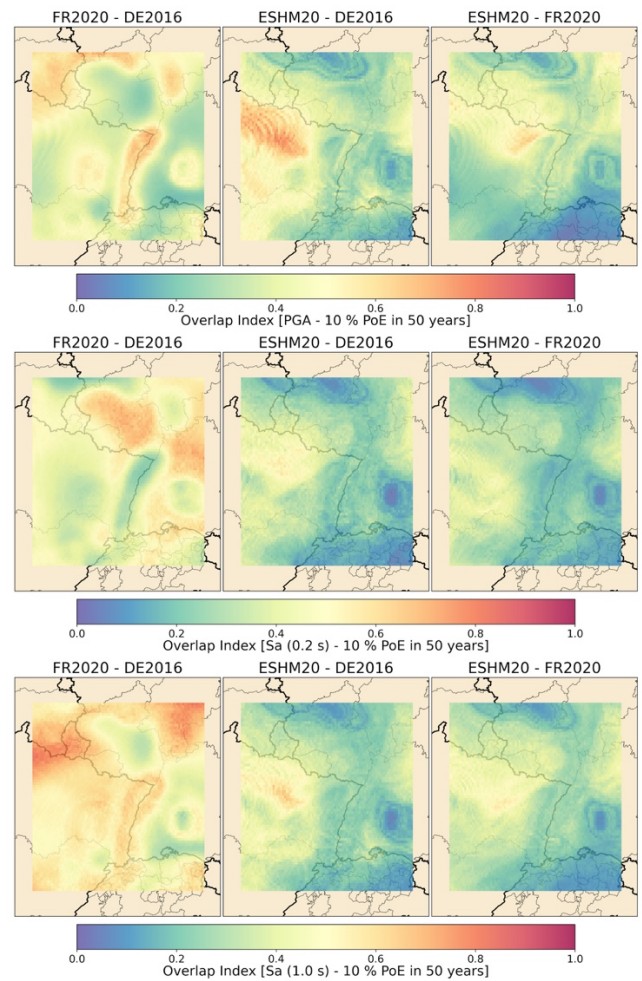

**Figure 20: As Figure19 considering the overlap index (OI)**

## 6. Discussion and Conclusion

The key aim of this study has been to set out a broader perspective on what we mean by *comparison* in the sense of PSHA models, and to illustrate different quantitative techniques to undertake this. With the development of the three models, we considered (FR2020, DE2016 and ESHM20), we are considering seismic hazard models that are sufficiently complex for "simple" difference maps to be an insufficient metric of comparison, yet at the same time the degree of complexity observed in the models is indicative of the current state of practice, particularly for PSHA in low-to-moderate seismicity regions. Model comparison, therefore, needs to account for this degree of complexity. In the current analysis we are considering three models developed by three separate teams of modellers, each of which was working for different objectives, with different tools and with a different geographical scope. Under such circumstances it is inevitable that the perspectives on seismic hazard that emerge for a common region (in this case the France/Germany border region) will display a degree of divergence, even if there





are some common elements in each of the models. These can reflect different views as to which uncertainties on parameters or components of the models should be captured by the logic tree and, depending on the tools available, how these uncertainties are evaluated. An important point often overlooked in model comparison is the extent to which the calculation software and its requirements and potential computational costs can influence the actual decisions made by the modeller. The execution of the epistemic uncertainty on the magnitude frequency relation in the three models is a clear example of the complex relationship

between tools and modelling decisions, and how these can lead to quite different outcomes.

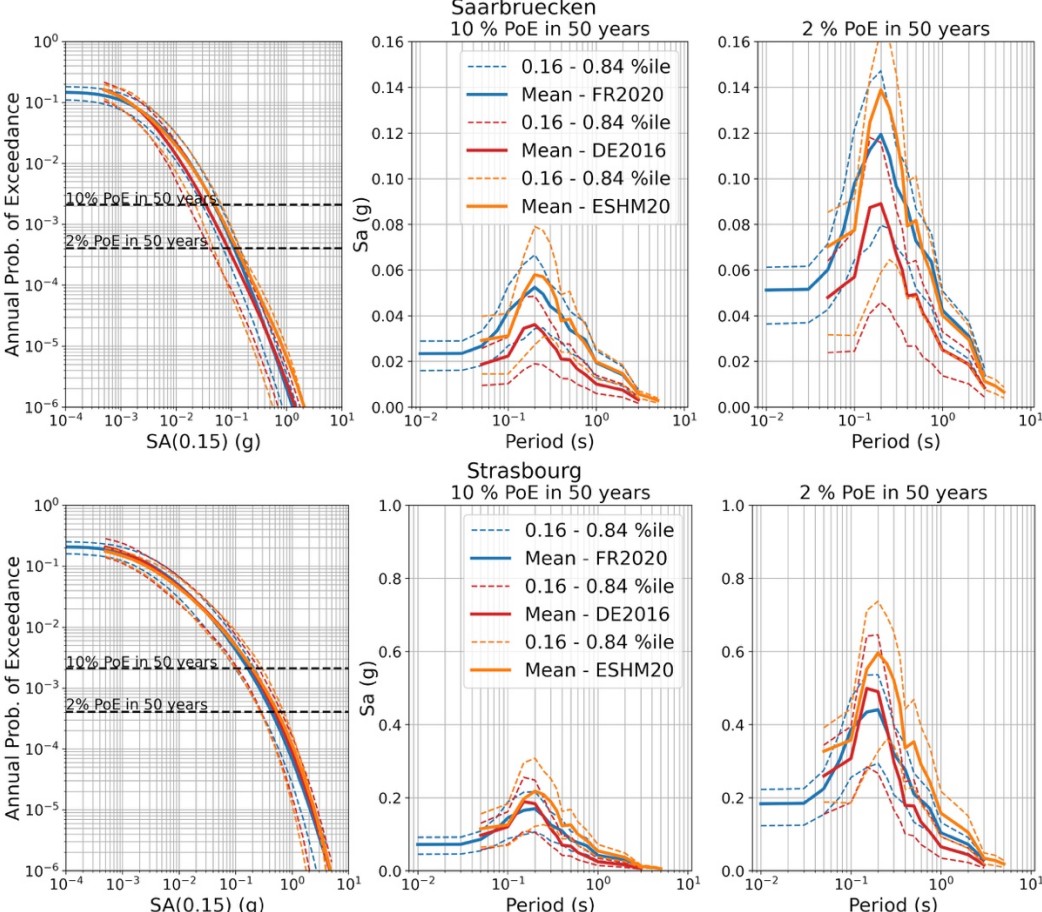

**Figure 21. Comparison of the distributions of seismic hazard for Saarbrücken (top row) and Strasbourg (bottom row) for hazard curves at Sa (0.15 s) (left) column) and UHS for 10 % PoE in 50 years (middle column) and 2 % PoE in 50 years (right column)**

To understand why and how PSHA models for a region diverge, one needs to break down the key factors in the model development and implementation process, and analyse each systematically: input data, modelling approach and philosophy, modelling tools, seismic hazard model components (e.g., seismogenic source model and ground motion model) and, finally, the seismic hazard model outcomes. The first two factors are compared more in a qualitative sense than a quantitative one. This is reflected in the presentation of the three models in section 2 of this paper, which juxtaposes the approaches the three





different models have taken to represent the seismogenic sources, to model the recurrence of earthquakes from each source
and to capture the expected ground motions from each earthquake. Each of the three models is working from input data that
shares many common characteristics such as the earthquake catalogue, which for ESHM20 comprises data from both the
FCAT-17 catalogue and the input seismic catalogue used by DE2016. Likewise, all three models had available geological
data for active faults in both the Upper and Lower Rhine Graben, and these have been either discarded, partly integrated or

fully integrated depending on the model. In terms of the modelling approach and philosophy, however, it is interesting to note
the many places in which the models have largely adopted a similar philosophy, yet the respective implementations yield
substantially different outcomes. A key example of this is the use of large-scale area zones (LASZs) based on tectonics and
smaller scale area zones based on local seismicity or geological features, both of which are balanced against a smoothed
seismicity model. The LASZs then form the prior zones, or direct measurements, for the MFRs of the small area source zones

within the maximum likelihood estimation, the outcomes of which are distributions of $a$ and $b$ values and their covariances.
Each model differs, however, in the specific zonations and in how the MFRs are, first, fit to the data and then how they are
mapped into branches of a logic tree.

One of the main opportunities that emerged from this work was to have all three models implemented in a common format for

use with the OpenQuake-engine seismic hazard and risk calculation software. This served several purposes, one of which being
to understand to what extent the three models differ by virtue of the calculation engine used to run them. The migration process
of a PSHA model from one software tool to another is seldom a trivial issue. Discrepancies emerge in computational
implementation of the PSHA calculations from one software to the next, which we separate into the following categories: i)
irreconcilable differences in operation, ii) bugs/errors and/or differences of interpretation, and iii) configurable parameters.

Migration of an implemented or existing model from one software to another is therefore a time-consuming process that
focuses on the finest details of the PSHA calculation rather than the general strategy for source and ground motion modelling.

Model migration differs from dual implementation, a practice becoming more widely adopted for quality assurance for critical
facilities that executes models in multiple software side by side, identifying discrepancies that are then discussed and

potentially resolved (e.g., Bommer et al., 2015; Aldama-Bustos et al., 2019). Migration assumes a reference seismic hazard
output from the original software, which the target software aims to reproduce regardless of whether the calculation processes
of the original software are deemed optimal. As perfect agreement in the calculations is rarely, if ever, achievable, we can only
define agreement between the implementations of a model in its original software and that in the target software in terms of a
degree of mismatch over a APoE range of relevance for application. We adopted ±10 % for APoE $\geq 10^{-4}$ (return period $\approx$

10,000 years) for this purpose, which applies firstly at a branch-by-branch level and then in terms of the mean and quantiles.
For DE2016 the target agreement was achieved for the mean and upper quantiles of seismic hazard for the vast majority of
sites considered and across multiple spectral periods. In some cases, the OpenQuake implementation estimated lower quantiles
that exceeded those of the original software beyond the specified target range. For FR2020 the target agreement could be





achieved for all area source branches individually; however, for the smoothed seismicity branches the OpenQuake hazard
curves appeared to be on average 20 – 30 % higher over the APoE range of interest. This resulted in OpenQuake's estimation
of the mean and quantiles to exceed those of the original software by between 10 – 20 % depending on the location and period,
which does not meet the target agreement. At the time of writing, no specific cause for this disagreement has been identified,
and we hope that this may still be resolved in subsequent iterations of the model.

The process of model migration for FR2020 and DE2016 was greatly facilitated in this case by the authors of the original
models, who supplied us with digital files of both the software inputs and the resulting seismic hazard curve outputs. Despite
this, each migration took several iterations, with more information regarding the calculation details needed as each discrepancy
is identified and, where possible, resolved. In both cases the specific details of the calculations were not found in the
accompanying documentation to the model and required clarification from the authors. In several cases the points of
clarification were not just related to small details of implementation but instead to major differences in how the models were
being executed within the calculation, sometimes even contradicting the supporting literature for the model. Though we are
sincerely grateful for the input of the model authors to aid this migration, this highlights a larger problem of model
reproducibility and a lack of standardisation in PSHA model documentation. A recommendation for improving practice here
is to require that where PSHA models are intended for use in large scale applications (e.g., a seismic design code) that the
digital input and output files for the calculation are made available and a standard documentation template developed that
requires the modellers to specify explicitly how the software they are using implements each component of the PSHA model,
which parts of the process are configurable and what values are adopted. Such information could greatly reduce the effort in
model migration and ensure greater transparency in the entire PSHA model implementation.

With the FR2020 and DE2016 models migrated to OpenQuake with a satisfactory level of agreement, we had a consistent
framework within which we can make quantitative comparisons of the hazard models, both in terms of the fundamental
components of the model inputs (i.e., the source and ground motion model) and the resulting hazard outputs. The latter describe
the extent to which models differ while the former provide insights as to why they do. The key issue we have sought to address
in the comparisons is the growing complexity of the logic trees that means we must now describe both the hazard model inputs
and outputs in terms of probability distributions and model space. This is the fundamental difference between the current
generation of PSHA models in Europe and many of their precedents. The logic trees of each of the three models considered
here incorporate not only alternative source models but multiple branches for epistemic uncertainty in the magnitude frequency
relation. This results in a much larger number of alternative predictions of activity rate and magnitude recurrence (400 for
FR2020 and 200 for DE2016), which begin to better resemble probability distributions (albeit of no specific functional form)
rather than individual alternative models. We have illustrated here how we can compare models in this context quantitatively,
first by looking at metrics describing the centre and variance of the distributions, then by invoking more information theoretic
metrics that quantify proximity of different probability distributions in model space, such as Kolmogorov-Smirnov Distance





$(D_{KS})$ and Wasserstein Distance ($D_{WS}$). Combining these different metrics and exploring their spatial trends can help provide insight as to where the models are most divergent, which can help identify future efforts could be placed to improve consistency
across models in future generations of seismic hazard models for Europe.

This last point takes us toward a critical question that we believe emerges from the work and affect how we may use the models in practice. *What can we do to effectively harmonise multiple seismic hazard models that cover a region*? This question is not necessarily a scientific one but rather a procedural one. Multiple groups developing separate models for a region and making
individual modelling decisions will inevitably result in different estimates of seismic hazard. This is widely recognised and procedures such as those adopted by SSHAC (US Nuclear Regulatory Commission, 2018) are intended specifically to formalise the management of information and scientific review in order to define the set of technically defensible interpretations and ensure that their centre, body and range are adequately represented. Seismic hazard modelling in Europe (illustrated here for the FR2020, DE2016 and ESHM20) does not currently take place within such a framework, as each model
has been commissioned for different purposes and by different organisations with no designation of a body to oversee coordination. ESHM20 aimed to integrate components of both the FR2020 and DE2016 models, yet practical limitations, the desire to incorporate new data and developments in PSHA, and the need to create a harmonised model at a larger scale, prevented it from faithfully incorporating all elements of the existing models into its framework. Divergence is therefore ensured from the very beginning of this process. Efforts such as the European Facility for Earthquake Hazard and Risk
(EFEHR) are seeking to provide a community structure to hazard and risk modelling around which data and tools are made openly available, and its working groups aim to focus on broadening the discussion of key issues and challenges for modelling. EFEHR cannot necessarily act in the role of technical integrator to the various organisations with remits to model hazard and risk in their respective countries, but it can and does provide harmonised data sets and tools for use as well open-source implementations of hazard and risk, all combined with extensive documentation. These can facilitate harmonisation from the
bottom up, eventually moving differences in modelling decisions, alternative interpretations and parameter uncertainties into a broad distribution of technically defensible interpretations across a region. We hope that if the EFEHR community is successful and can continue to expand, divergence between the models may eventually be minimised to better reflect the actual epistemic uncertainty in a region.

**Supplementary Material**

Additional information relating to the France (Drouet et al., 2020) and Germany (Grünthal et al., 2018) PSHA models and their implementation into OpenQuake is available with the online version. These include images and information about the model translation (Appendix A) and comparisons of the seismic hazard results for the respective countries and selected cities (Appendix B).




**Author Contribution**

Author contributions are defined according to the CRediT Taxonomy. **Conceptualization:** GW, FC, GD, IZ. **Methodology and Investigation:** GW, PI, CB. **Writing (original draft):** GW. **Data Curation and Software:** GW, CB, PI, **Writing –**
**Review and Editing:** GW, FC, GD, IZ, PI, CB. **Funding Acquisition:** FC, GD, IZ

**Declaration of Competing Interests**

The authors declare no conflicts of interest.

**Acknowledgements**

The work presented here has benefitted from discussion with, and data provided by, Emmanuel Viallet, Stephane Drouet, David Baumont and Gabriele Ameri (FR2020), Gottfried Grünthal and Dietrich Stromeyer (DE2020), and Laurentiu Danciu (ESHM20) and members of the European Facility for Earthquake Hazard and Risk (EFEHR). We thank Sum Mak (formerly of GFZ) for his proposal and explanation of the Overlap Index. Seismic hazard calculations have been undertaken using the OpenQuake-engine, for which we thank the GEM Foundation for their ongoing development and support. Other quantitative
analysis used tools from the Python Scientific Stack (including Numpy, Scipy, Pandas, Matplotlib, H5Py and Geopandas), and some maps have been prepared using QGIS. This work has been supported by the SIGMA2 research program and partially funded by Électricité de France.

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
