# Peer review of "Strategies for Comparison of Modern Probabilistic Seismic Hazard Models and Insights from the Germany and France Border Region"

_Natural Hazards and Earth System Sciences, 2023_

## Referee Comment (RC3)

**"Strategies for Comparison of Modern Probabilistic Seismic Hazard Models and Insights from the Germany and France Border Region"**

**Revision**

Weatherill et al.,2023 focuses on PSHA models for France and Germany, alongside the 2020 European Seismic Hazard Model, to investigate the differences in model components and highlight the challenges and strategy for harmonising the different models into a common PSHA calculation software. Recent significant earthquakes with immense economic and human loss stated once more the importance of seismic hazard assessment models. Weatherill et al.,2023 presented an important work about the detailed comparison of three Central European hazard models. The manuscript was an interesting one to read. The presented research contributes to understanding of the effects regarding the selected model parameters in PSHA.

Manuscript represent an excellent substantial contribution to the understanding of natural hazards and their consequences. scientific and/or technical approaches and the applied methods are good enough to be published. The results discussed in an appropriate and balanced way in detail. The scientific data, results and conclusions were well presented in a clear, concise, and well-structured way

Some comments about the work are listed below:

Other seismic hazard models in globe should be mentioned in "Introduction" with a comment on the similarities and differences with Central European Hazard models.

A preliminary paragraph can be presented with more detail for the reason in selecting these three models and the region.

Other models from Greece, Portugal, Spain beside ESHM13 (in Lines 49-51) may be commented.

Gutenberg, B. and Richter, C.F. (1944), may be referred as the base for understanding seismicity rates, which is a fundamental component of probabilistic seismic hazard analysis.

Basilic et al., 2013. (The European Database of Seismogenic Faults (EDSF) compiled in the framework of the Project SHARE https://seismofaults.eu/edsf13) may be commented/referred.

"Considerable degree of divergence in the tectonic zonations" should be explained commented in Line 261.

Discussions on using the Complete logic tree with differences in France and Germany should take place.

Line 200: … is not quite  clear…

Line 552: The fundamental framework for PSHA is largely unchanged changed ?

---

## Referee Comment (RC4)

**Review of the manuscript NHESS-2023-98**

The paper NHESS-2023-98 "Strategies for Comparison of Modern Probabilistic Seismic Hazard Models and Insights from the Germany and France Border Region" presents the quantitative comparison of three complex, independent seismic hazard models and accompanying maps at national and regional scales using metrics from informatic theory. The aim of this paper is of great interest to the seismic hazard community because so far the comparison of seismic hazard models with overlapping regions has been limited to the comparison of hazard curves for sites located in the same territory between the models. Instead, quantitative tools to compare the individual components of the hazard models have not been published yet as far as I know. I really appreciate the highlight of the reproducibility and transparency of seismic hazard models estimated using different software packages in the conclusion. This is one of the biggest limitations in comparing different PSHA studies for the same region.

There are also a few adjustments, which could improve the manuscripts.

1) The manuscript attempts to explain very complex seismic hazard models in a reasonable length for an article and therefore some sentences are very difficult to follow. I would suggest simplifying the long sentences perhaps breaking them into two parts for better readability.

2) The introduction discusses the increasing complexity of seismic hazard models from the first and second generations (lines 34-35 and 72) but there are no citations of such models at national scales. To make this point clearer I would suggest to include one or two examples of first- and second-generation national hazard models for the same country.

3) In Section 2 the three models should be described in chronological order for the year of the publication starting with Germany, then France and ESHM20. Furthermore, the same features should be described for each model. For example, the earthquake catalogue is described for the FR2020 but not for DE2016 and ESHM20; the GMM component is described for DE2020 but not for DE2016 and ESHM20. The same order for the three models should be followed throughout the manuscript for both the text and the figures. For example, in Figure 5 the plots in the first column should be DE2016, second column for FR2020, and third column for ESHM20. It will help the reader to navigate through a manuscript that discusses many elements.

4) I find it surprising that the earthquake catalogues, including strategies to homogenise the magnitude scale, decluster the catalogue, and assess the catalogue completeness, used by the three models are not discussed in more detail, except for a little mention in Section 2.3. Since the source models, especially the recurrence statistics of the three models, and how they differ are extensively described in the manuscript, the earthquake catalogues used to compute the recurrence parameters are a reason to explain the difference in the seismic hazard models. For this reason, they should be described better.

Below there are a few (technical or editorial) comments on the manuscript.

Lines 34-35: Provide references for "several successive generations of seismic hazard models" and "multiple models".

Lines 50-51: I would suggest ordering the national seismic hazard models in chronological order.

Line 52: CEN (2004) is not listed in the reference section.

Line 66: The acronym "GMMs" does not correspond to ground motion model components. Should it be "ground motion model (GMM) components"?

Line 71: "The increased in sophistication…" seems not correct.

Line 74: The acronym PoE is not explained.

Line 83: The word "and" between "proprietary software" and "into the open-source" should be deleted.

Line 108: A word, probably "that" is missing between "Germany (DE2016 hereafter)" and "was prepared". Also, the acronym DIBt does not correspond to Deutsches Institut für Bautechnik, should the "t" be deleted?

Line 109: DIN 4149 is not listed in the reference section.

Line 120: It seems that $S\alpha$ and $S\beta$ are not explained.

Table 1: What does "315 (West) / 5985 (East)" indicate? Are east and west related to the region, i.e. roughly East and West Europe?

Lines 159-160: In the sentence "large scale area zones delineating tectonically based 160 domains ("Grands Domaines")" some wording is missing.

Lines 169-170: It is difficult to understand how "the smoothed seismicity branches differ in approach between FR2020 and DE2016". Is there a specific reference for the zoneless approach in FR2020? Similarly, for the comparison of the zoneless model for DE2016 and ESHM20, the reference of Helmstetter and Werner (2012) should be included in lines 169-170.

Line 184: Add the references for "existing models from Belgium, Switzerland, and the United Kingdom".

Line 210: If the comparison between the smoothed seismicity models is done earlier (i.e. lines 169-170) this reference should be also cited earlier (see also previous comment). Alternatively, such a comparison ("The smoothed seismicity branches differ in approach from those found in both FR2020 and ESHM20, as DE2016 uses an adaptive kernel with magnitude-dependent 170 bandwidth based on the method of Woo (1996).") can be moved here.

Line 246: The ";" does not seem the correct punctuation. Perhaps it can be replaced with ". This is". In any case, this sentence is very long and could be simplified by dividing it into 2 (or even 3) parts.

Lines 162-163: The word "define" is repeated twice in the same sentence ("one another in defining three zones of similar extent that define the Paris Basin,"). A synonym could replace one of them.

Line 284: Is there anything missing in "… of with *dm*.."?

Line 285: Add a comma between "(0,$\sigma$(m)" and "the epistemic…".

Line 287: The citation "Miller and Rice (1983)" is indicated as "Miller & Rice (1983)" elsewhere in the manuscript. I would suggest consistency in the notation and check the use of "and" and "&" throughout the manuscript when the citation consists of two names only.

Line 300: "Kagen" should be replaced with "Kagan".

Figure 5: It would be helpful to include a legend for the size of the circles of the top plots. What is the minimum magnitude for the earthquakes plotted here? 4.5 Mw? Is the colour scale in the bottom right-hand side plots applied to the three plots? and therefore do the branches for the FR2020 have the same weight?

Line 402: Add a comma between "(FR2020)" and "a hydrid".

Line 404: (USNRC, 2012) is not listed in the reference section.

Line 430: Is the sentence "The latter is fit to NGA West 2 data but using a simpler functional form than the NGA West 2 GMMs, which more suited for the level of parameterization commonly found in moderate to low seismicity regions" related to Bindi et al. (2017)? If so, I would suggest change "(as explained), and Bindi et al. (2017). The latter is fit" with "(as explained). Bindi et al. (2017) is fit"

Figure 9: Is there a reason why 5.25 and 6.50 Mw were chosen for the earthquake scenarios, and not 5 and 6 Mw for example?

Line 539: Pagani et al. (2014) is not listed in the reference section. I assume this citation corresponds to Pagani et al., 2014a or 2014b.

Line 555: Assatorians & Atkinson, 2014 is not listed in the reference section.

Line 579: Add "packages" between "software" and "characterise".

Line 592: The words "highlighted emphasized" are redundant, one of them should be deleted.

Line 604: Allen et al., 2020 and Abbot et al., 2020 should be listed in alphabetic order since the year of the publication is the same.

Lines 693-694: It seems that the word "that" is missing between "representation" and "allows".

Lines 698-702: This sentence is difficult to follow. I would suggest rephrasing it to improve readability.

Figure 12: If I have understood correctly, the gridded activity rates plotted in Figure 12 account for the areal source models, the smoothed source models, and the fault source models for DE2016, FR2020, and ESHM20. Is that correct?

Figure 12: Why is North Germany white in the middle and bottom row plots? Is the activity rate very low to be white? It seems there is an abrupt change from yellowish and white.

Line 745: Perhaps, the word "of" between "similarity" and "dissimilarity" should be replaced with "or".

Lines 747-748: Include references for the weighted Kolmogorov-Smirnov Statistic and Wasserstein Distance.

Figure 17: Do the extreme values in the colour scale correspond to the minimum and maximum values in the maps?

Line 822: The authors should explain what "acceleration level A" is because I think it was not mentioned before.

Line 836: Replace "difference" with "different" at the end of this line.

Lines 849-850: Include the coordinates used for the cities of Saarbrücken and Strasbourg.

Lines 873: The word "consider" was repeated twice in the same sentence ("…we considered, we are considering seismic hazard models that are sufficiently…"). I would suggest using synonyms for one of them.

Lines 871-879: Some sentences here are quite long and perhaps they could be simplified.

Lines 943-949: These two sentences could be rephrased for better readability. Furthermore, the word "that" is repeated three times in the same sentences.

Line 971: US Nuclear Regulatory Commission, 2018 is not listed in the reference section.

Reference section: The references Goulet et al. (2021), Vilanova et al. (2014), and Weatherill and Cotton (2020) are included here but are not cited in the text. Also, the reference Meletti, D'Amico and Martinelli does not include the year (I assume it is 2013).

---

## Author Response (AR1)

**Strategies for Comparison of Modern Probabilistic Seismic Hazard Models and Insights from the Germany and France Border Region**

Graeme Weatherill[1], Fabrice Cotton[1,4], Guillaume Daniel[2], Irmela Zentner[3], Pablo Iturrieta[1,4], Christian Bosse[1]

[1]GFZ German Research Centre for Geosciences, Potsdam, 14473, Germany
[2]Électricité de France, Aix-en-Provence, 13290, France
[3]Électricité de France, EDF R & D Lab Paris-Saclay, 91120, France
[4]University of Potsdam, Potsdam-Golm, 14476, Germany

*Correspondence to*: graeme.weatherill@gfz-potsdam.de

**REPLY TO REVIEWER COMMENTS**

A comprehensive set of replies to the reviewer comments and indication of the major changes to the manuscript can be found in the online materials of the paper (https://nhess.copernicus.org/preprints/nhess-2023-98/) under the "Discussion" tab.

We sincerely thank Peter Powers, Ilaria Mosca and two anonymous reviewers for their thorough reviews and constructive comments that have helped to improve the quality of the revised manuscript.

---

## Author Response (AR2)

**Strategies for Comparison of Modern Probabilistic Seismic Hazard Models and Insights from the Germany and France Border Region**

Graeme Weatherill[1], Fabrice Cotton[1,4], Guillaume Daniel[2], Irmela Zentner[3], Pablo Iturrieta[1,4], Christian Bosse[1]

[1]GFZ German Research Centre for Geosciences, Potsdam, 14473, Germany
[2]Électricité de France, Aix-en-Provence, 13290, France
[3]Électricité de France, EDF R & D Lab Paris-Saclay, 91120, France
[4]University of Potsdam, Potsdam-Golm, 14476, Germany

*Correspondence to*: graeme.weatherill@gfz-potsdam.de

**REPLY TO EDITOR COMMENTS**

We submit the revised manuscript following comments made by the editor. Our reply to these comments is found below.

*Dear authors,*

*I have reviewed your revised manuscript, and am pleased to inform you that the revised version of your manuscript can be accepted for publication subject to a minor revisions in our journal Natural Hazards and Earth System Sciences (NHESS). I would like to request a few minor (and some technical) revision before publication.*

We thank the editor for their review and comments on the manuscript. We have endeavored to address the editors comments as best we could in the revised manuscript.

*Minor revisions:*

*(1) I agree with Reviewer 4 that it would be helpful to include a legend for the size of the circles of the top plots in Figure 5. If this clutters the figure, please add it as text to the figure caption, indicating that the minimum magnitude for the earthquakes is Mw 3.0, and that the color scale in the bottom right-hand site plots applies to all three plots (i.e., the branches for FR2020 have the same weight). Similarly, please add a line or two explaining why 5.25 and 6.50 Mw events were chosen for the earthquake scenarios in Figure 9. This can be done in the manuscript text or as part of the figure caption.*

A legend for the circle size has now been added to the figure in question. Likewise, a couple of sentences explaining the selection of the scenarios has been added where the comparisons of the ground motion models are first introduced.

*(2) Please label the geographical names you use in the text (e.g., Upper/Lower Rhine Graben, The Rhine, Alpine Foreland, Albstadt shear zone) and place names such as Luxembourg, Karlsruhe, Freiburg im Breisgau, and Basel. Not all readers are familiar with these locations and will struggle finding the regions you mention in the text. You could do this on one of the existing maps (figure 4 or 5) or create a separate map to avoid cluttering.*

A new figure (now Figure 1) has been added to the introduction section that shows the region in question and labels the major tectonic features and locations mentioned in this article.

*(3) Please define 'active faults' and what you mean by 'long-term activity rates for the faults'.*

While the seismogenic fault source compilations mentioned in the respective models have not adopted a single standard definition of "active faults", we include here a definition based on the information and parameterization regarding a fault that would be needed in order for it to be included as a seismogenic source in the PSHA models. The meaning of long-term activity rates has also been clarified.

*Editorial comments/General editorial comment.*

We have addressed the editorial comments in the revised manuscript.

*I look forward to receiving the revised version of your manuscript.*

*Solmaz Mohadjer*

*NHESS Editor*